# TEST-TIME SCALING VIA METRIC GEOMETRY FOR LLM REASONING

## ABSTRACT

Test-Time Scaling (TTS) methods improve the reasoning capability of large language models (LLMs) by generating multiple independent Chain-of-Thoughts (CoTs) and aggregating them via designed policies. Despite effective, this ensemble approach incurs expensive inference costs due to the repeated model calls. In this paper, we propose a physics-inspired framework that achieves the accuracy gains of multi-calls TTS within a single or few LLM calls. It conceptualizes LLM reasoning as navigating through a Maze, a complex puzzle through which one has to find a path to achieve specific goals. The proposed *Maze* paradigm embeds candidate exemplars and domain knowledge into a multiplex latent manifold and learns a high-dimensional metric space. In inference, *Maze* metrics can identify a single or a few optimal paths; each path refers to an ordered sequence of exemplars, forming a few-shot prompt that guides the LLM to the correct answer. Empirically, in reasoning benchmarks including GPQA, MMLU-pro, GSM8K, MATH-500, and AIME, *Maze* matches or exceeds the accuracy of the Best-of-$N$ strategies while reducing the computational cost by 60∼80%. These results support *Maze* to be a principled geometric alternative to brute-force TTS, enabling low-latency, interpretable, and computation-efficient reasoning for complex tasks. We also advocate for an interesting width-depth equivalence in LLM reasoning under the *Maze* framework: any solution achievable by many shallow trials can be attained by a suitably planned sequence of reasoning steps.

## 1 INTRODUCTION

Large language models (LLMs) increasingly rely on Test-time Scaling (TTS) to elicit reasoning, from Chain-of-Thought (CoT) prompting and self-consistency voting to explicit search over intermediate thoughts (Wei et al., 2022; Wang et al., 2022; Yao et al., 2023; Besta et al., 2024). Although these strategies can improve accuracy, they typically require many LLM calls, which increases latency and computational cost. For example, self-consistency typically requires generating a large number $N$ of reasoning paths to ensure that the correct answer dominates the vote (Chen et al., 2023). More importantly, recent analysis shows that adding LLM calls can yield non-monotonic returns due to correlated errors and heterogeneous query (Chen et al., 2024). Such brute-force "width-first search" (many shallow attempts) is inefficient and infeasible for very large $N$, encountering a fundamental scalability problem in test-time reasoning.

In parallel, a complementary line of work demonstrates that the choice and order of few-shot exemplars[1] is a first-order factor for in-context learning, *i.e.*, swapping or reordering the same exemplars can significantly affect performance(Lu et al., 2021; Guo et al., 2024). This has spurred retrieval-based demonstration selection and bandit-style or flatness-aware prompt optimization (Rubin et al., 2021; Shen et al., 2023; Wu et al., 2023), yet most approaches either (i) keep the LLM in the inference loop to score candidates at test time, (ii) treat content and order largely as separable heuristics, or (iii) ignore the layer-wise geometry of hidden states that governs how prompts are transported through the model. In short, TTS strategies expend width (parallel shallow attempts) to gain reliability, while fine-designed prompting expends depth (sequential search steps), and both can be costly and hard to interpret in different ways.

---

[1]An exemplar is a labeled example (an input-output pair) used to demonstrate a task to a model. We can simply view an exemplar as a training sample written in the form of a question-label pair.

In this paper, by investigating *width* as selection over many candidate paths versus *depth* as repeated sampling per query, we propose the *Maze*[2] to reconcile width and depth in LLM reasoning. *Maze* is a test-time planner that replaces multi-call sampling with LLM-free path selection over an ordered pool of exemplars. *Maze* builds multilayered manifolds on the hidden layers of an LLM, learns a lightweight energy functional that jointly captures content and order effects, and then selects a maximum energy "path" (ordered prompt) using only the cached representations of the query, issuing a single LLM call to answer. In particular, *Maze* explicitly accounts for the ordering and diversity of the exemplars, related to known sensitivities in prompt design (Lu et al., 2021; Cegin et al., 2024). By learning a metric that captures these effects, *Maze* reduces the reliance on large ensembles or a fine-designed prompt tuning.

The design is motivated by the intrinsic geometry of the neural representations, where the semantic structure is concentrated in specific layers and the intrinsic dimensionality is low (Valeriani et al., 2023). Moreover, selection theory under Lipschitz reward assumptions suggests that compact projections can preserve reward ordering over large candidate sets (Magureanu et al., 2014). In parallel, recent work (Pei & Wang, 2023; Pei et al., 2024b) has shown that importing *dynamical* and *geometric* structure into representation learning can produce compact but strong models. Unlike inference-based retrieval and bandit-based prompt optimization that tunes prompts via expensive trial-and-error, *e.g.*, using reinforcement learning or local geometric or variational semantics to find an optimal global ordering (Deng et al., 2022; Wu et al., 2023; Shen et al., 2023; Pei et al., 2024a), *Maze* externalizes scoring offline, eliminates LLM-in-the-loop ranking and targets the tail of high-value paths, offering a principled alternative to compute scaling via vote-and-aggregate systems whose returns may saturate or decline (Wu et al., 2023; Chen et al., 2024).

We further hypothesize and theoretically support a fundamental Width-Depth Conservation principle in LLM reasoning. Informally, we conjecture that for any reasoning task that can be reliably solved by a large ensemble of $N$ shallow CoT trials, there exists an equivalently successful solution using far fewer calls (even a single call) organized as a deeper, well-planned reasoning trajectory. In other words, any "wide" exploration with many independent attempts can be converted, without loss of correctness, to a "deep" exploration with a smaller number of dependent sequential steps, and vise versa. This hypothesis formalizes an intuitive trade-off in problem-solving resources and suggests that brute-force voting is not the only route to high reliability.

Empirically, we validate *Maze* in a suite of challenging reasoning benchmarks without any task-specific prompt tuning. In experiments on MMLU-Pro (Wang et al., 2024), GPQA (Rein et al., 2024), GSM8K (Cobbe et al., 2021), MATH-500 (Hendrycks et al., 2021), and AIME (Veeraboina, 2023), a single *Maze* exploration often matches or exceeds the accuracy of much heavier baselines, such as 32-sample self-consistency voting, despite using an order-of-magnitude fewer LLM calls. For example, using a model with 4B∼14B parameters such as Qwen3 (Yang et al., 2025), Mistral (Jiang et al., 2023), and LLaMA3 (AI@Meta, 2024), our one-call planner Maze achieves comparable results to an optimal Best-of-16 ensemble, reducing inference latency and cost by 60∼80%. These findings empirically confirm the proposed width-to-depth conversion: carefully chosen deep prompts can replace massive shallow ensembles. In summary, our contributions are threefold.

- We introduce Maze, a test-time reasoning paradigm that transforms unguided trial-and-error into a single guided path, offering a new alternative for prompt engineering and inference-time optimization.
- We develop a theoretical analysis of width–depth equivalence in LLM reasoning, showing that a task solvable by many independent CoT samples is also solvable by an appropriate sequential prompting strategy with far fewer calls.
- We provide extensive empirical validation across different tasks and model architectures, *e.g.*, Qwen3 4B/8B, Mistral 7B, LLaMA3 8B, showing that *Maze* achieves strong performance with significantly fewer inference calls.

Our work aims to suggest a new TTS paradigm for reasoning with LLMs, *i.e.*, to leverage strategically planned depth instead of brute-force width. See Appendix C.2 for more motivation and theoretical analysis on methodology design. See Appendix D for more related work.

---

[2]A Maze is a puzzle-like structure consisting of paths, corridors, walls, and obstacles with an objective of finding an optimal path from a starting point to the ending point.

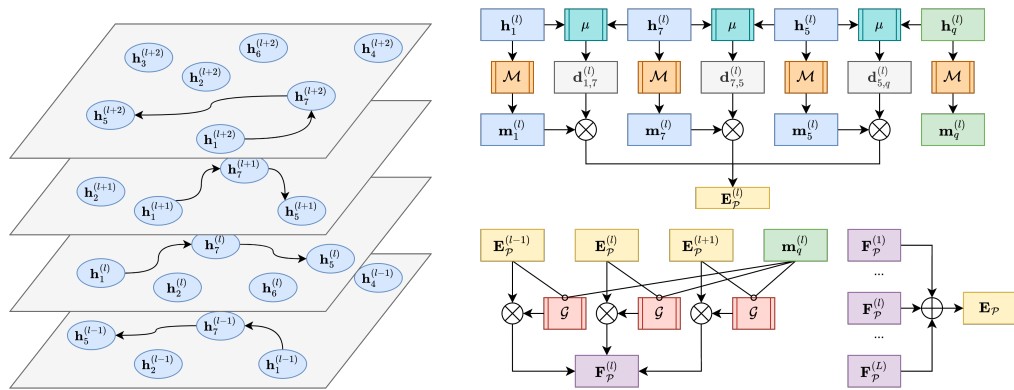

(a) *Maze* path as many layer-wise sub-paths                (b) Aggregating layer-wise *Maze* sub-paths

Figure 1: (**a**) **Layer-wise paths in a multi-layered latent manifold.** Given an LLM, queries and QA exemplars are embedded and cached as layer-wise hidden states $\mathbf{h}^{(l)}$ of each layer $l$. *Maze* treats neural layers as a multi-layered manifold where nodes are exemplars and queries, connected by edges measured with intra-layer and cross-layer asymmetric metrics. A path refers to an ordered few-shot prompt, *i.e.*, system prompt $\Rightarrow$ QA exemplars $\Rightarrow$ test query. (**b**) **Single-call LLM-free path planning**. We learn lightweight global mappings, metric $\mu$, mass mapping $\mathcal{M}$, and inter-layer fusion $\mathcal{G}$, so that the path energy $\mathbf{E}_{\mathcal{P}}$ reliably ranks each candidate path $\mathcal{P}$. At test time, we (i) cache once the test query's hidden states $\mathbf{h}_q^{(l)}$, (ii) evaluate layer-wise energies $\mathbf{E}_{\mathcal{P}}^{(l)}$ and overall energy $\mathbf{E}_{\mathcal{P}}$ for each path $\mathcal{P}$ within a large paths pool without calling an LLM, (iii) pick the maximum-energy path, and (iv) invoke the LLM once to produce the answer. Multi-call Maze, *a.k.a.*, *MultiMaze*, uses the same planner that returns the top-$k$ paths followed by $k$ LLM calls with Best-of-$k$ strategies.

## 2 METHODOLOGY: MAZE-GUIDED REASONING

### 2.1 REASONING AS A *Maze* PATH

We conceptualize the LLM reasoning process as navigating through a "maze", *i.e.*, a complex structure designed as a puzzle through which one has to find a path to achieve specific goals. The accessible path in the *Maze* corresponds to the QA exemplars, and the wall on the sides of the path represents "knowledge" induced by an LLM. The start of the *Maze* is the system prompt (or the first exemplar), and the end is the test problem. Solving a complex problem then amounts to finding appropriate intermediate QA exemplars in a specific order to form a path (few shot prompts). The end of this ordered path gains sufficient "energy" to guide the LLM to answer the problem correctly.

Formally, for a fixed LLM and a candidate pool $\mathcal{E} = \{(q_i, a_i)\}$ of QA exemplars, our goal is to build an external latent space, *i.e.*, the *Maze* equipped with metrics that can rank sets and orders of exemplars for a test problem $q$. Then, during inference, *Maze* can select a small, ordered few-shot prompt that maximizes the chance that the LLM answers $q$ correctly, while using minimal extra test time computation. Before inference, for each exemplar $e_i = (q_i, a_i)$, run the fixed LLM once and capture the hidden states of $L$ chosen layers in the LLM.

For fair comparison, we use the reasoning LLM itself as the encoder to extract the hierarchical encoding of a path. When direct access to an LLM's weights is limited, we can also use a relatively lightweight open-sourced LLM (*e.g.*, Qwen3-4B) as the encoder and the performance gain remains almost unchanged, empirically; we will explore more in Sec. 4.5. To encode a path, let $\mathbf{h}_i^{(l)} \in \mathbb{R}^d$ be the average hidden states captured in the layer $l$ of the LLM acting as the encoder; collect $H(e) = \{\mathbf{h}_i^{(1)}, ..., \mathbf{h}_i^{(L)}\}$ be the embedding group of the exemplar $e$. Do the same for each test query $q$ during inference. We view each layer $l$ as a "room" and define intra-layer and cross-layer distances between exemplars. Distances are computed through learnable metrics, trained offline on inference samples that capture the correctness of the LLM under specific few-shot prompts. We then fuse these distances with structure-aware terms to obtain a single overall energy for each *Maze* path.

The energy associated with each path reflects the domain-specific knowledge that the LLM can extract through that path. A higher energy value indicates that the LLM, when using the exemplars from this path as a few-shot prompt, is more likely to acquire richer and more relevant knowledge for answering the test query. Consequently, the probability of the LLM correctly solving the query increases. This mechanism arises because the LLM can, in effect, assess its own prior inference performance on the task, thereby evaluating the adequacy of its task-related knowledge and reasoning capabilities. In essence, this mechanism can be viewed as an implicit form of self-reflective reasoning. The LLM leverages its own history of inference performance to gauge the alignment between its knowledge and the demands of the current task, supporting a process akin to self-assessment and self-improvement.

Overall, a path $\mathcal{P}$ is simply an ordered few-shot prompt: system $\Rightarrow K$ exemplars $\Rightarrow$ test query. A layer $l$ refers to a particular neural layer of the encoder/LLM. For each exemplar and the test query, we extract layer-wise hidden states $h(l)$, producing a layer-wise view of the same exemplar sequence. We then score candidate paths by aggregating these multi-layer views via learned intra-layer and cross-layer metrics and a fusion function, yielding a single scalar energy or score for ranking. The constructed Maze structure serves as a scaffold that enables the LLM to consolidate its own historical inference behaviors and refine its understanding of the current task. Unlike prior approaches that rely on heavyweight reward models, our Maze structure provides a lightweight, offline knowledge summarization mechanism. Using a set of trainable, interrelated, and extensible metrics as a substitute for standalone reward models, it enables more efficient testing time scaling.

## 2.2 INTRA-LAYER AND CROSS-LAYER METRICS

We now discuss how to assign a metric to a path. As in Fig. 1a, a Maze path can be viewed as a cluster of $L$ sub-paths, each associated with one of the $L$ neural layers. Consequently, we introduce two families of metric functions: *Intra-layer* metric that measures the distances between the nodes within the same layer (one per layer, $L$ in total); *Cross-layer* metric that measures the distances between nodes across different layers (with the ordering taken into account, $L^2$ in total).

For brevity, we denote intra-layer metrics as $\mu^{(l)}$ and cross-layer metrics as $\mu^{(l,l^*)}$, both defined as mappings $\mathbb{R}^d \times \mathbb{R}^d \mapsto \mathbb{R}^r$ with $r < d$, where $d$ is the hidden dimension in LLM and $r$ is the *metric dimension*. The distance $\mathbf{d}_{x,y}^{(l)} \in \mathbb{R}^r$ between two exemplars $e_x$ and $e_y$ in layer $l$ is:

$$\mathbf{d}_{x,y}^{(l)} = \mu^{(l)}\big(\mathbf{h}_x^{(l)}, \mathbf{h}_y^{(l)}\big) + \sigma \circ \sum_{w=1}^{W} \mu^{(l\pm w,\,l)}\big(\mathbf{h}_x^{(l\pm w)}, \mathbf{h}_y^{(l)}\big) \tag{1}$$

where $\sigma$ is a nonlinear operator such as *Tanh*, and we set $W = 2$ for a better computational trade-off. Intuitively, $\mathbf{d}_{x,y}^{(l)}$ aggregates direct similarity within the layer $l$ and the influences of other layers. Note that the metric is an asymmetric metric, *i.e.*, $\mu(a,b) \neq \mu(b,a)$. In addition, the metric should reflect only the relation between $e_x$ and $e_y$, not their absolute representations.

## 2.3 MAZE ENERGY AND FUSION

The path energy should reflect the effort required to move through the path, depending on both the path and the node-specific properties, *e.g.*, *mass*. We assume that the *mass* information for each exemplar has been encoded in its hidden state. To extract this, we define the mass mapping $\mathcal{M} : \mathbb{R}^d \mapsto \mathbb{R}^r$, where $r < d$ is identical to the metric dimension. Here, we use $r$-dimensional *mass* instead of a scalar because we demand a richer latent representation. For clarity, we use $\mathbf{m}_x^{(l)} = \mathcal{M}\big(\mathbf{h}_x^{(l)}\big)$. Taking a $K$-shot prompt consisting of exemplars $e_1, \ldots, e_K$, with $e_0$ the system prompt and $e_{K+1}$ the test problem, we have a path $\mathcal{P} = \{e_0, \ldots, e_{K+1}\}$ consisting of $K + 2$ nodes. In layer $l$, the *static* path energy $\mathbf{E}_{\mathcal{P}}^{(l)}$ is:

$$\mathbf{E}_{\mathcal{P}}^{(l)} = \sum_{i=0}^{K} \sigma\big(\mathbf{m}_i^{(l)} \cdot \mathbf{d}_{i,i+1}^{(l)}\big). \tag{2}$$

As in physics, *work* associated with each directed edge between nodes is relevant to the product of node mass and inter-node distance. To aggregate across all layers while incorporating the relevance

of the test problem, we define the *dynamical* path energy as a fusion term $\mathbf{F}_{\mathcal{P}}^{(l)}$:

$$\mathbf{F}_{\mathcal{P}}^{(l)} = \mathcal{G}\big(\mathbf{m}_{K+1}^{(l)}, \mathbf{E}_{\mathcal{P}}^{(l)}\big) \cdot \mathbf{E}_{\mathcal{P}}^{(l)} + \sigma \circ \sum_{w=1}^{W} \mathcal{G}\big(\mathbf{m}_{K+1}^{(l)}, \mathbf{E}_{\mathcal{P}}^{(l\pm w)}\big) \cdot \mathbf{E}_{\mathcal{P}}^{(l\pm w)} \tag{3}$$

where the fusion mapping $\mathcal{G} : \mathbb{R}^r \times \mathbb{R}^r \mapsto \mathbb{R}^r$ quantifies communications between neural layers and we set $W = 2$ for better computation-accuracy trade-off. Then the overall path energy is

$$\mathbf{E}_{\mathcal{P}} = \sum_{l \in \mathcal{L}} \mathbf{F}_{\mathcal{P}}^{(l)} + \sum_{l^* \notin \mathcal{L}} \mathcal{G}\big(\mathbf{m}_{K+1}^{(l^*)}, \mathbf{E}_{\mathcal{P}}^{(l^*)}\big) \cdot \mathbf{E}_{\mathcal{P}}^{(l^*)}. \tag{4}$$

where $\mathcal{L} = \{1+W, 2+W, ..., L-W\}$ used to avoid the border case. This yields a generalized path energy for a few-shot prompt. The learnable mappings are as follows.

$$\{\mu^{(l)}, \mu^{(l,l^*)}, \mathcal{M}, \mathcal{G}\}, \quad l \neq l^* \text{ and } l, l^* \in \{1, \ldots, L\} \tag{5}$$

These mappings determine distances, masses, and inter-layer relevance, and can be instantiated with flexible nonlinear structures.

## 2.4 LEARNABLE MAPPINGS FOR *Maze*

By Theorem 1, if the mappings declared in Eq. 5 are sufficiently expressive, the path energy computed from a single LLM call can approximate the evaluation score of a verifier or reward model in a Best-of-$N$ (BoN) strategy, which otherwise requires $N$ LLM calls.

In fact, we do not expect that a single call strategy will outperform BoN. Instead, the goal is to achieve a better efficiency–performance trade-off: approaching the reasoning ability of BoN while drastically reducing the inference cost. Therefore, we can simply adopt lightweight linear projections as the basic instantiations, balancing expressivity with computational efficiency. For the layer-dependent metrics, we simplify them as a single global metric, *i.e.*, $\mu^{(l)} = \mu^{(l,l^*)} = \mu$:

$$\mu\Big(\mathbf{h}_x^{(l)}, \mathbf{h}_y^{(l^*)}\Big) = (\mathbf{h}_x^{(l)} - \mathbf{h}_y^{(l^*)}) \cdot \mathbf{W}_\mu , \tag{6}$$

where $\mathbf{W}_\mu \in \mathbb{R}^{d \times r}$ is a trainable linear projection. For the mass mapping, we apply

$$\mathcal{M}(\mathbf{h}_i^{(l)}) = \mathbf{h}_i^{(l)} \cdot \mathbf{W}_{\mathcal{M}} , \tag{7}$$

where $\mathbf{W}_{\mathcal{M}} \in \mathbb{R}^{d \times r}$ is a trainable linear projection. For the fusion mapping $\mathcal{G}$, we simply employ a multilayer perceptron (MLP) as

$$\mathcal{G}(\mathbf{m}, \mathbf{E}) = \sigma(\text{concat}(\mathbf{m}; \mathbf{E}) \cdot \mathbf{W}_{\mathcal{G}}^{(hid)}) \cdot \mathbf{W}_{\mathcal{G}}^{(out)}, \tag{8}$$

where $\mathcal{G}^{(hid)} \in \mathbb{R}^{2r \times h}$ and $\mathbf{W}_{\mathcal{G}}^{(out)} \in \mathbb{R}^{h \times r}$ are trainable linear projections with $h > r$.

Intuitively, we can view the path energy as the cost of moving a physical object along a track. Each exemplar acts like a checkpoint with its own mass, while the metric defines the resistance or distance between checkpoints. The total energy then reflects how much "effort" is required to traverse the entire path from the system prompt to the test problem. In this analogy, intra-layer and cross-layer metrics capture both local friction and cross-track influences, while $\mathcal{G}$ functions as a controller that decides which tracks matter most to solve the final task.

## 2.5 LEARNING THE *Maze* OFFLINE

We learn the mappings $\{\mu^{(l)}, \mu^{(l,l^*)}, \mathcal{M}, \mathcal{G}\}$ so that the scalar Maze energy $\mathbf{E}_{\mathcal{P}}$ of an ordered $K$-shot path $\mathcal{P}$ is a reliable, compute-light approximation for downstream correctness under a fixed LLM $F$. Training is entirely offline: we (i) log results of $F$ on budgeted few-shot prompts, (ii) construct contrastive positive/negative prompts by single local edits (content replacement or order swap), and (iii) fit the *Maze* so that correct prompts have higher energy.

Let $\mathcal{E} = \{e_i = (q_i, a_i)\}_{i=1}^{N}$ be the QA exemplars of the training set. For each training query $q$, we sample dozens of $K$-shot base prompts $\{(S, \pi)\}$ with $S \subset \mathcal{E}$ and order $\pi$. Note that $S$ should not include the exemplar containing $q$. We run $F$ once per base prompt and record the

correctness $r(q, S, \pi) \in \{0, 1\}$. To make training informative and fine-grained, for each base prompt we generate minimal edits: (i) content edit at position $j$: replace $e_j \in S$ by $\tilde{e} \in \mathcal{E} \setminus S$; (ii) order edit: swap two positions $j \leftrightarrow k$. We accept an edited prompt only if it flips the outcome on $q$ from correct to incorrect or vise versa. Thus, for each $q$ we collect contrastive pairs:

$$\left(\mathcal{P}^+(q), , \mathcal{P}^-(q)\right) \quad \text{s.t.} \quad \mathcal{P}^- \text{ differs from } \mathcal{P}^+ \text{ by one local edit and } r(\mathcal{P}^+) > r(\mathcal{P}^-). \tag{9}$$

We handle stable-easy and stable-hard queries by forcing label flips via a short sequence of edits until we obtain a valid pair or reach the maximum attempts. Empirically, $50k$ contrastive pairs built from around 10k base prompts are sufficient to train a well-formed Maze. The LLM hidden states are pre-cached; training involves only MLP and projection updates. Before inference, we cache the hidden states in layerwise $H(e) = \{\mathbf{h}^{(l)}(e)\}_{l=1}^{L}$ for all exemplars $e \in \mathcal{E}$ and $H(q)$ for all training $q$. Given $\mathcal{P} = (e_0, \ldots, e_{K+1})$ with $e_{K+1} \equiv q$ and $e_0$ the system prompt, we evaluate the per-layer energy $E_{\mathcal{P}}^{(l)}$ and the fused scalar energy $E_{\mathcal{P}}$ as specified in Section 2.

We choose Adam (Kingma, 2014) as the optimizer and train the Maze mappings by directly supervising the energy as a score and imposing pairwise margins on contrastive edits. In summary, offline learning shapes the *Maze* so that energy tracks correctness; the correct paths become short and low-resistance in the learned geometry.

### 2.6 TEST-TIME DEPLOYMENT AND MULTI-CALLS *Maze*

**Single-call *Maze*.** At test time, given a test query $q$, we compute $H(q)$ once. Then, we sample a modest number $M$ of candidate $K$-shot prompts, evaluate their energies $\{E_{\mathcal{P}_m}\}_{m=1}^{M}$ without calling the LLM, and pick the maximum energy prompt. Finally, we call $F$ once with that prompt to produce the answer. Because energy evaluation is a small number of matrix–vector operations, we can safely let $M$ be large (*e.g.*, $10^3$) while keeping the test-time computation dominated by a single LLM call.

**Multi-calls *Maze*.** We can also pick the top-$k$ maximum energy prompts and employ $k$ LLM calls to generate $k$ responses followed by a TTS policy, *e.g.*, Best-of-$k$ or majority vote. This multi-calls *Maze*, *a.k.a*, *MultiMaze*, is supposed to have better reasoning capability than the single-call *Maze*, as well as worse computational efficiency. Actually, the multi-calls *Maze* is compatible with other TTS policies as we can consider it as an offline verifier implemented without an LLM call.

## 3 CAN LLM-FREE PATH PLANNING EMULATE TEST-TIME SCALING?

We propose that a single-call *Maze* over ordered exemplar sets equals Best-of-$N$ strategies. The informal statement of this result is as follows.

> Under mild assumptions on generalization, the single-call path planner approximates the high-probability correctness as the *Best-of-$N$* strategies, while requiring only one LLM invocation plus low-cost metric evaluations.

See Theorem 1 in Appendix A.1 for a formal statement and the proof.

We also conclude a width-depth conservation hypothesis for LLM reasoning and its weaker formalization. These arguments make LLM-free path planning a possible principled low test-time cost surrogate for width-heavy test-time scaling. The informal statement is as follows.

> Any reasoning task whose solution can be reliably obtained via a large number $N$ of shallow independent LLM calls (large width and small depth) is convertible, is often reducible (empirically validated across our benchmarks) into an equivalent problem solvable with a smaller number $M \ll N$ of deeper, strategically planned LLM calls (small width and large depth).

This width–depth conservation hypothesis formalizes an intuitive trade-off in LLM reasoning: the width of sampling via multi-agent policies can be "transferred" into the depth of a single carefully planned LLM call. This formulation provides a unifying mathematical lens through which Maze's transformation of multiple shallow LLM calls into fewer deep calls can be understood as a special

Table 1: **Overall accuracy (%) on LLM reasoning tasks with single-call methods.** Original uses one random few-shot call. pFLAT (Shen et al., 2023) and EASE (Wu et al., 2023) are geometric and ordering-aware exemplar optimization methods. *Maze* selects the optimal path from the randomly generated pool and uses a single LLM call.

|  |  | MMLU-pro | GPQA | GSM8K | MATH500 | AIME 21-24 |
|---|---|---|---|---|---|---|
| Mistral-7B | Original | 24.29 | 29.69 | 53.46 | 42.12 | 7.78 |
|  | pFlat | 26.43 | 31.70 | 58.08 | 45.28 | 12.35 |
|  | EASE | 26.79 | 32.81 | 58.46 | 46.12 | 13.68 |
|  | Maze (Ours) | **28.57** | **33.93** | **62.69** | **47.90** | **18.56** |
| LLaMA3-8B | Original | 41.37 | 31.65 | 80.90 | 48.40 | 4.12 |
|  | pFlat | 44.50 | 35.80 | 85.78 | 51.36 | 9.20 |
|  | EASE | 45.10 | 35.85 | 85.91 | 52.80 | 9.35 |
|  | Maze (Ours) | **48.43** | **39.17** | **87.57** | **53.90** | **14.66** |
| Qwen3-4B | Original | 49.11 | 36.50 | 89.86 | 84.46 | 39.81 |
|  | pFlat | 51.00 | 40.18 | 92.14 | 85.16 | 43.25 |
|  | EASE | 51.32 | 39.99 | 91.60 | 85.10 | 45.32 |
|  | Maze (Ours) | **54.13** | **43.18** | **92.67** | **87.28** | **47.32** |
| Qwen3-8B | Original | 59.74 | 38.35 | 89.98 | 86.82 | 47.57 |
|  | pFlat | 66.13 | 49.95 | 92.03 | 88.24 | 49.25 |
|  | EASE | 66.95 | 48.30 | 91.77 | 87.98 | 50.32 |
|  | Maze (Ours) | **70.03** | **53.25** | **93.06** | **90.16** | **53.23** |

case of a wider conservation law in problem solving resources. A formal statement is presented in the Appendix A.2 as a weaker Theorem 2 with a proof sketch.

The low-rank structure and the low intrinsic dimensionality of LLM fine-tuning or representations justify Theorem 1's subspace model. Moreover, the Johnson-Lindenstrauss Lemma and the Universal Approximation Theorem guarantee the existence of compact and accurate *Maze* energies. With appropriately small projection/MLP widths, evaluating thousands of candidate paths is cheaper than a single forward pass of modern LLMs. Hence, LLM-free path planning can emulate (and often outperform in compute/latency) BoN-style test-time scaling while using one call in the high-margin regime, and few calls otherwise.

## 4 EXPERIMENTS

### 4.1 DATASETS AND BASELINES

We adopt five widely used reasoning benchmarks that span knowledge, scientific problems, and competition-level mathematics: MMLU-Pro (professional knowledge QA; multi-choice, single answer) (Wang et al., 2024), GPQA (graduate-level multi-discipline QA; multi-choice) (Rein et al., 2024), GSM8K (grade-school math word problems; free-form, numerical final answer) (Cobbe et al., 2021), MATH-500 (subset of competition math; free-form with final numeric/symbolic answer) (Hendrycks et al., 2021), and AIME (Veeraboina, 2023), which we use the 2021-2024 split for the test set, and others for the train set.

These tasks are intentionally diverse to test the cross-domain generalization of a single *Maze* prompt and learned metric. We evaluate exact answer accuracy for multi-choice tasks and exact match (EM) for numeric or symbolic answers on reasoning tasks. We used frozen LLMs in sizes 4B to 14B, including Qwen3-4B/8B/14B, Mistral-7B and LLaMA3-8B; no fine-tuning is performed. For baselines, we compare majority-vote (Chen et al., 2024), oracle-guided BoN, pFLAT (Shen et al., 2023), and EASE (Wu et al., 2023). We omit the verifier-guided methods because they are theoretically upper-bounded by the optimal (oracle-guided) BoN method.

We primarily focus on accuracy/exact match to evaluate the performance. We also emphasize that evaluating energies for thousands of candidate paths is negligible compared to a single LLM call (subsecond on our hardware), so the compute is dominated by the number of LLM invocations, *i.e.*, the computational cost is explicitly measured by No.LLM calls.

Table 2: **Overall accuracy (%) on reasoning tasks with multiple-call methods.** *Majority vote* uses 32 independent CoT samples with voting. The *Optimal BoN* assumes an oracle-guided "reward model" that, among $N = 16$ calls, returns correct if at least two samples are correct. *MultiMaze* selects the top-16 paths by energy and applies the same oracle-guided BoN (also $N = 16$ calls). Computing energies for all paths takes $< 1$ sec., which is negligible to an LLM call (10-30 sec.).

|  |  | MMLU-pro | GPQA | GSM8K | MATH500 | AIME 21-24 |
|---|---|---|---|---|---|---|
| Mistral-7B | Majority vote | 28.93 | 37.28 | 73.46 | 48.25 | 19.25 |
|  | Optimal BoN | 35.00 | 39.96 | 80.38 | 52.67 | 23.46 |
|  | MultiMaze (Ours) | **37.50** | **43.30** | **84.62** | **55.44** | **27.43** |
| LLaMA3-8B | Majority vote | 53.53 | 41.03 | 93.05 | 54.28 | 15.65 |
|  | Optimal BoN | 60.59 | 46.15 | 94.59 | 56.76 | 17.82 |
|  | MultiMaze (Ours) | **62.35** | **51.54** | **95.37** | **60.30** | **21.43** |
| Qwen3-8B | Majority vote | 82.35 | 52.63 | 93.75 | 89.22 | 55.62 |
|  | Optimal BoN | 85.71 | 63.16 | 95.10 | 92.35 | 60.10 |
|  | MultiMaze (Ours) | **86.55** | **68.95** | **95.86** | **93.46** | **63.35** |
| Qwen3-14B | Majority vote | 82.80 | 54.32 | 94.65 | 92.30 | 55.12 |
|  | Optimal BoN | 85.98 | 65.27 | 95.72 | 92.90 | 59.39 |
|  | MultiMaze (Ours) | **87.55** | **70.92** | **95.98** | **94.95** | **61.32** |

## 4.2 IMPLEMENTATION DETAILS

***Maze* training (offline).** We implement the learnable mappings with lightweight linear projections and a small MLP as specified in section 2.4, and train with Adam with a learning rate around $1e-3$ on contrastive prompt pairs produced by single local edits (content replacement or order swap) until correctness flips (Section 2.5). By default, we set the metric dimension $r = 64$ and the hidden dimension $h = 96$. The resulting No.Params is around 0.35M. We use $K \in \{4, 8\}$ for the $K$-shot setting; for larger $K$, we apply a two-stage curriculum that first emphasizes content edits in $K/2$. No gradients flow through the base LLM. Note that we train the *Maze* mappings using the contrastive pairs obtained from all of the benchmarks, *i.e.*, MMLU-pro, GPQA, GSM8k, MATH-500, each benchmark counts for an equal split. The total number of contrastive pairs ranges from 10,000 to 500,000. We will explore more about how the training size affects the performance.

**Inference.** For a test query $q$, we cache hidden states once, evaluating the energies of the path pool. The path pool is made up of $M$ randomly generated paths with a size around $10^2$; we will explore how $M$ affects performance later. For *Maze*, we choose the maximum energy path from the path pool and perform one LLM call. For *MultiMaze*, we choose the highest $k$ maximum energy paths and make $k$ calls with the majority vote. Empirically, energy evaluation is matrix-vector dominated and far cheaper than one LLM forward, enabling large path pools without latency overhead.

**Unified decoding & prompting.** All methods use the same system prompt, max tokens (4096), and temperature (0.1); baselines and *Maze* share the same candidate exemplar pool to ensure fairness. We use a GeForce RTX 4090 GPU with 24GB memory, and the total time required to train a set of metric models typically ranges between several hours. The total time to generate the training set, *i.e.*, contrastive pairs, ranges between several days. We only need to generate the training set once.

## 4.3 EVALUATION ON REASONING TASKS

Table 1 reports the single-call results. In all reported settings, Maze consistently outperforms other single-call prompting methods like pFLAT and EASE, with notable gains in reasoning benchmarks. Unlike EASE that trains a neural bandit to search global order for all queries, Maze trains a geometry once and performs LLM-free scoring per query, enabling single-call parity. Table 2 presents the multi-call results. *MultiMaze* with 16-call consistently outperforms 32-call self-consistency and, in some cases, approaches or exceeds the optimal BoN oracle on the same budget while keeping the call count 50% lower than the 32-sample BoN. We also validate Accuracy vs. No.LLM calls in *MultiMaze* and observe that a 3-call *MultiMaze* sits close to a 10-call majority vote and a 6-call Oracle-guided BoN on GPQA with Qwen3-4B (Appendix Figs. 5c–5d). Finally, consistent with recent compound inference scaling analyses (Chen et al., 2024), we observe that a naive increase in

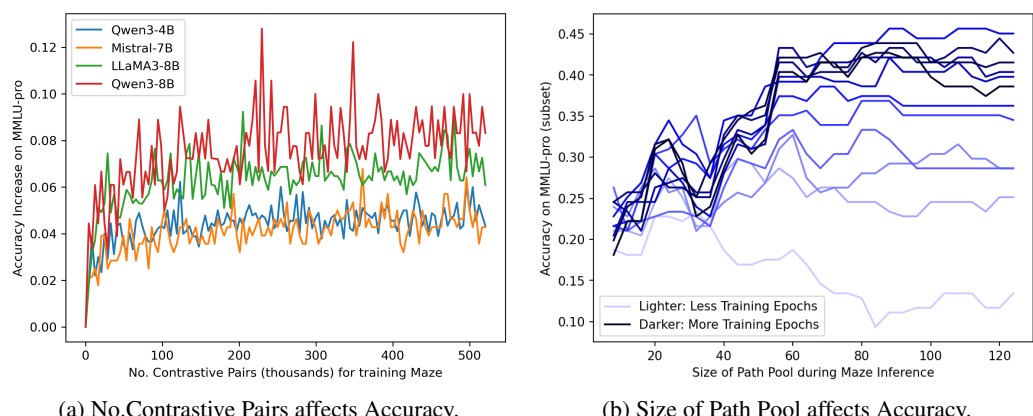

(a) No.Contrastive Pairs affects Accuracy.  (b) Size of Path Pool affects Accuracy.

Figure 2: **Scaling behavior of *Maze*.** (**a**) Accuracy @ MMLU-pro vs. number of contrastive pairs used to train the metric mappings. (**b**) Qwen3-4B's Accuracy @ MMLU-pro vs. path-pool size $M$ at inference. Darker hues indicate more training epochs (from 2 to 32) for the same configuration.

the number of LLM calls may not be uniformly beneficial; *Maze* selection mitigates redundancy and correlation between failures by targeting high value paths. Additional ablations, including exemplar content vs. ordering, the number of contrastive pairs used to train *Maze*'s metric, layer selection, path-pool size, metric capacity, and qualitative visualizations, are provided in Appendix B.1 to B.9.

### 4.4  EFFECT OF EXEMPLAR CONTENT AND ORDER IN A MAZE PATH

Both the exemplar *content* and their *ordering* within a Maze path, *i.e.*, the few-shot prompt materially affect performance. For Qwen3-4B on MMLU-Pro, we take each *Maze*-selected optimal path and generate three variants by shuffleing the exemplar order while keeping the content fixed. The accuracy drops by 2.8% from 54.1% to 51.3% on average. Without *Maze* at all, the accuracy is 49.1%, *i.e.*, lower by 5% than with *Maze*. Hence, for this setting, the contribution attributable to *ordering* is approximately 57% of the total *Maze*-induced gain. In other LLMs and datasets, we observe order contributions in the range of 43% to 76%. In general, this result validates that the optimal paths selected with *Maze* are both *content-* and *order*-sensitive.

### 4.5  DEPLOYMENT WITH A PROXY ENCODER FOR BLACK-BOX LLMS

*Maze* requires no gradient, and does not intervene the generation process or intermediate representation. The only model-side signal used is a fixed hidden-state embeddings for exemplars and queries; importantly, this encoder is swappable. In practice, we cache the exemplar embeddings using a lightweight shared global encoder and train *Maze*'s metric externally, then use the resulting planner to select the few-shot prompts for an LLM without any access to the LLM's internal weights. Empirically, using Qwen3-4B as a universal encoder for all LLMs (including LLaMA and Mistral) preserves performance gains (Table 4). *Maze* also applies to closed-source models such as GPT-5 and still produces consistent improvements over both single-call and multi-call baselines (Table 5).

## 5  CONCLUSION

We introduced *Maze*, a geometry-driven test-time scaling framework that converts width-heavy ensembles into a single strategically planned traversal through an exemplar space. Under mild low-rank and margin assumptions, a one-call planner *Maze* recovers the BoN ranking while offering a lower inference cost. Empirically, *Maze* and its small-call variant match or surpass strong TTS baselines across knowledge and math reasoning benchmarks, with consistent gains in latency and token efficiency. Future work will focus on scalability to larger-LLM setting and tighten the theory frame.

ETHICS STATEMENT

We state that this submission raises no question regarding the Code of Ethics.

REPRODUCIBILITY STATEMENT

We upload the sample code for extracting hidden states from an LLM and building a *Maze* as supplementary material. The basic configuration and training scripts for *Maze* are also included. The baselineckbone LLMs and the benchmarks along with the datasets are both publicly available.

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

# A  APPENDIX: MATHEMATICAL VALIDATION

## A.1  *Maze* EMULATION OF BEST-OF-N UNDER LOW-RANK AND MARGIN

**Theorem 1** (*Maze* Emulation of BoN under Low-Rank + Margin). *Fix an LLM denoted as $F$ and a candidate pool $\mathcal{P} = \{(S, \pi)\}$ of ordered $K$-shot prompts for a query $q$. Let $R : \mathcal{P} \mapsto [0, 1]$ denote the BoN reward for each prompt, i.e., the success probability of $F$ when using $(S, \pi)$ (expectation over decoding randomness). We assume:*

- *Low-rank representation: for each $(S, \pi)$, let $z(S, \pi) \in \mathbb{R}^D$ with $D = Ld$ be the concatenation of the $L$ layer-wise hidden states of $q$ or exemplars. There exists an $r$-dimensional subspace $U \subset \mathbb{R}^D$ with $r \ll D$ such that every $z(S, \pi)$ has its low-rank alternative in $U$;*

- *Lipschitz reward: the reward $R$ is $L_R$-Lipschitz on the compact set $Z = \{z(S, \pi)\} \subset U$ with respect to Euclidean distance;*

- *Margin: let $(S^*, \pi^*) = \arg\max_{(S,\pi)} R(S, \pi)$. We assume that there exists a positive scalar called gap $\Delta = R(S^*, \pi^*) - \max_{(S,\pi)\neq(S^*,\pi^*)} R(S, \pi) > 0$.*

*Then we claim that for any $\varepsilon \in (0, \Delta/2)$ and failure probability $\delta \in (0, 1)$, there exists:*

- *a Johnson-Lindenstrauss projection $\Pi : \mathbb{R}^D \mapsto \mathbb{R}^m$ with $m = O(\frac{r}{\varepsilon^2} \log \frac{|\mathcal{P}|}{\delta})$, and*

- *parameters $\theta$ of the Maze mappings $\{\mu^{(\cdot)}, \mathcal{M}, \mathcal{G}\}$ implemented as linear projections and a one-hidden-layer MLP,*

*such that, with probability at least $1 - \delta$ over $\Pi$, the linear-algebraically computed Maze energy $\mathbf{E}_\theta(S, \pi)$ uniformly $\varepsilon$-approximates $R(S, \pi)$ on $\mathcal{P}$: $\forall (S, \pi) \in \mathcal{P} : |\mathbf{E}_\theta(S, \pi) - R(S, \pi)| \leq \varepsilon$, and hence preserves the maximizers: $\arg\max_{(S,\pi)} R(S, \pi) = \arg\min_{(S,\pi)} \mathbf{E}_\theta(S, \pi)$.*

*Proof.* We sketch the proof in the following four steps:

*Step 1 (Subspace embedding).* Because $\mathcal{Z} \subset U$ with $\dim(U) = r$, there exists a random linear map $\Pi$ to $\mathbb{R}^m$ with $m = \mathcal{O}\left(\frac{r}{\varepsilon^2} \log \frac{|\mathcal{P}|}{\delta}\right)$ that is an $\varepsilon$-distortion embedding for all pairwise distances on $\mathcal{Z}$ with probability $\geq 1 - \delta$. Thus, for any $z, z' \in \mathcal{Z}$, $|(z - z')| - \varepsilon \leq |\Pi z - \Pi z'| \leq (1 + \varepsilon)|z - z'|$. In particular, composing $R$ with $\Pi^{-1}$ over $\Pi(\mathcal{Z})$ keeps $L_R$ finite; we absorb the distortion into the Lipschitz constant.

*Step 2 (Uniform function approximation).* The set $\Pi(\mathcal{Z})$ is compact. By universal approximation with non-polynomial activation, a one-hidden-layer MLP $\varphi_\theta : \mathbb{R}^m \to \mathbb{R}$ uniformly approximates $R \circ \Pi^{-1}$ in $\Pi(\mathcal{Z})$ within error $\varepsilon/2$.

*Step 3 (Maze parameterization).* The Maze energy $E_\theta$ is an MLP applied to linear features constructed from pairwise differences and masses. Since linear maps followed by concatenation are affine reparameterizations of $\Pi z(S, \pi)$, we can implement $\varphi_\theta$ within the *Maze* architecture by choosing $\mu, M, G$ appropriately, resulting in $|E_\theta - R \circ \Pi^{-1}| \leq \varepsilon/2$ on $\Pi(\mathcal{Z})$.

*Step 4 (Uniform error and argmax stability).* Combining Steps 1–3 gives a uniform bound $\sup_{(S,\pi)} |E_\theta(S, \pi) - R(S, \pi)| \leq \varepsilon$. If $\varepsilon < \Delta/2$, then for any suboptimal $(S, \pi)$,

$$E_\theta(S, \pi) \geq R(S, \pi) - \varepsilon > R(S^*, \pi^*) - \Delta - \varepsilon \geq R(S^*, \pi^*) - 2\varepsilon > E_\theta(S^*, \pi^*), \quad (10)$$

so the maximizer of $E_\theta$ coincides with the maximizer of $R$. □

**Complexity.** Evaluating $E_\theta$ over $M$ candidates costs $\mathcal{O}\big(M \cdot (LK, m + H)\big)$ arithmetic operations, where $LK, m$ comes from linear projections of layer-wise states for $K$ exemplars and $H$ is the hidden size of the MLP. With $m = \tilde{\mathcal{O}}(r/\varepsilon^2)$ and small $H$, this is negligible compared to one forward call of $F$.

□

## A.2 THE WIDTH–DEPTH CONSERVATION HYPOTHESIS IN MATH REASONING

**Definition A.1** (Reasoning Tree). *A reasoning tree for a math problem is a rooted ordered tree whose nodes correspond to intermediate assertions (lemmas, subgoals, or steps) and whose edges correspond to inference steps. The* width $W$ *of the tree is the maximum number of children of any node, and the* depth $D$ *is the length (in edges) of the longest root-to-leaf path.*

**Definition A.2** (Resource Budget). *Given a reasoning tree of width $W$ and depth $D$, its* resource budget *is defined as $R(W, D) = W \times D$, interpreted as the total capacity of the 'thought step' available.*

**Theorem 2** (Weak and restricted Width–Depth Conservation). *For any math problem $\mathcal{P}$, let $R^*_{\min}(\mathcal{P})$ be the minimum resource budget required to solve $\mathcal{P}$ using any reasoning tree. Then for any pair $(W, D)$ satisfying $W \cdot D \geq R^*_{\min}(\mathcal{P})$, there exists a reasoning tree of width at most $W$ and depth at most $D$ that solves $\mathcal{P}$, and conversely, no tree with $W \cdot D < R^*_{\min}(\mathcal{P})$ can solve $\mathcal{P}$.*

*Proof Sketch.* For any math problem $\mathcal{P}$, let $R^*_{\min}(\mathcal{P})$ be the minimal resource budget (Definition A.2) over all reasoning trees (Definition A.1) that solve $\mathcal{P}$. Then Theorem 2 holds. We must show two directions:

**(i) Sufficiency:** If $W \cdot D \geq R^*_{\min}(\mathcal{P})$, then there is a width-$W$, depth-$D$ tree solving $\mathcal{P}$.

By the definition of $R^*_{\min}$, there exists some tree $T^*$ of width $W^*$ and depth $D^*$ that solves $\mathcal{P}$ with $W^*D^* = R^*_{\min}$. If $W^* \leq W$ and $D^* \leq D$ then $T^*$ already satisfies the width or depth constraints. Otherwise, apply the lemma 6 to $T^*$ to obtain a chain of length $L \leq W^*D^* = R^*_{\min}$. Factor $L$ as $L = W \cdot D$ (which is possible since $W D \geq R^*_{\min}$) and then apply the lemma 7 to reconstruct a tree of width $\leq W$ and depth $\leq D$ that still solves $\mathcal{P}$.

**(ii) Necessity:** If $W \cdot D < R^*_{\min}(\mathcal{P})$, then no tree of width $W$ and depth $D$ can solve $\mathcal{P}$.

This follows immediately from the definition of $R^*_{\min}$: By minimality, any solving tree must have budget $W'D' \geq R^*_{\min}$, so if $W D < R^*_{\min}$ no such $(W, D)$ bounded tree can exist.

Combining (i) and (ii) completes the proof of Theorem 2. $\square$

## A.3 AUXILIARY LEMMAS

**Lemma 3** (Metric Generalization). *Suppose $\mu_\theta$ is trained by minimizing a margin-based hinge loss of margin $\Delta > 0$ on $m$ positive/negative path pairs. Then with probability at least $1 - \delta$, for all held-out correct paths $\pi^+$ and incorrect paths $\pi^-$,*

$$D_\theta(\pi^-) - D_\theta(\pi^+) \geq \Delta - \varepsilon, \qquad \varepsilon = O\left(\sqrt{\frac{\log(1/\delta)}{m}}\right).$$

*Proof Sketch.* By standard Rademacher complexity bounds for margin losses, the empirical margin in the $m$ training pairs concentrates to within $O(\varepsilon)$ of its population expectation. A union bound over all held-out pairs yields the claim. $\square$

**Lemma 4** (Coupon-Collector Coverage). *Let there be $Q$ equally likely "coupons" (distinct multisets of size $N$). Drawing $L$ coupons ensures that every coupon appears at least once with probability at least $1 - \eta$, provided*

$$L \geq Q \ln \frac{Q}{\eta} = O(N \ln(N/\eta)) \quad \text{(since } Q = \Theta(N) \text{ here).}$$

*Proof Sketch.* The classical coupon-collector argument shows that $Q \ln Q + Q \ln(1/\eta)$ draws suffice to see all $Q$ coupons with failure prob. $\leq \eta$. In our case $Q = O(N)$ since there are $M$ distinct CoT outputs and $N$ multisets, so $Q = \binom{M+N-1}{N} = O(N)$ for fixed $M$. $\square$

**Lemma 5** (Chernoff Concentration for Best-of-$N$). *If a single CoT sample is correct with probability $p > 0.5$, then the majority vote over $N$ independent samples is correct with probability*

$$p_N \geq 1 - \exp\left(-2N(p - 0.5)^2\right).$$

*Proof Sketch.* Apply the multiplicative Chernoff bound to the sum of independent $\{0, 1\}$ correctness indicators. $\qquad\square$

**Lemma 6** (Sequentialization of Parallel Reasoning). *Let $T$ be a reasoning tree of width $W$ and depth $D$. Then there exists an equivalent (i.e., derives the same final conclusion) chain of inferences, i.e., a tree of width $1$, of length at most $W \cdot D$.*

*Proof.* Number the nodes of $T$ in breadth-first order. Whenever a node has multiple children, we serialize its $k \leq W$ inferences into a linear sequence of $k$ steps. Repeating this for each level of $T$ incurs at most $W$ steps per original level, and there are $D$ levels, giving a total length $\leq W \cdot D$. This chain preserves correctness since each serialized inference is identical to its parallel counterpart. $\qquad\square$

**Lemma 7** (Parallelization of Sequential Reasoning). *Let $C$ be a chain of inferences of length $L$. Then for any factorization $L = W \cdot D$, there exists a reasoning tree of width at most $W$ and depth at most $D$ that derives the same conclusion in $L$ total inference-steps.*

*Proof.* Partition the linear chain $C$ into $D$ contiguous blocks each of size at most $W$ (since $W \cdot D = L$). At the root, we apply the first block of up to $W$ inferences in parallel (making the root have up to $W$ children), then recurse on each child with the next block, and so on. This constructs a tree of depth $D$ and width at most $W$ whose leaves correspond to the final assertion in $C$. $\qquad\square$

# B    APPENDIX: MORE EMPIRICAL RESULTS

## B.1    ABLATION STUDY

We ablate the components of Maze's metric-geometry to isolate where gains arise (Table 3).

**Layer usage**. Using only the last or only a middle layer to define the manifold underperforms the all-layers multiplex variant. This confirms that different layers contribute complementary signals for path selection and that a multiplex geometry is preferable to a single-slice manifold.

**Cross-layer couplings**. Removing cross-layer metrics in the distance (Eq. 1) or cross-layer fusion in the dynamic energy (Eq. 8) degrades accuracy. Thus, asymmetric, order-aware interactions between layers are critical: they let the planner exploit how ordering interacts with representation transport across layers, improving the discriminability of near-tie paths.

**Supervision signal**. Replacing contrastive local edits (content replacement or order swaps that flip correctness) with random pairs consistently hurts performance. Contrastive supervision concentrates learning on decision boundaries of correctness, producing a sharper energy landscape for ranking candidates.

**Overall**. The complete *Maze*, with multiplex metrics, cross-layer fusion, and contrastive supervision, delivers the largest gains across datasets and backbones, indicating that performance is not driven by a single heuristic, but by the coherent geometric design of the energy functional.

Table 3: **Ablation study for single-call *Maze* on LLM reasoning tasks.**

|  |  | MMLU-pro | GPQA | GSM-8k | MATH500 |
|---|---|---|---|---|---|
| LLaMA3-8B | Original | 41.37 | 31.65 | 76.95 | 48.40 |
|  | only last layer | 44.7 | 35.8 | 77.6 | 51.28 |
|  | only middle layer | 46.1 | 36.8 | 78.1 | 52.16 |
|  | w/o Cross-layer metrics | 46.6 | 37.2 | 78.3 | 51.26 |
|  | w/o Cross-layer fusions | 45.7 | 36.9 | 79.1 | 50.80 |
|  | random pairs | 45.2 | 36.4 | 78.3 | 50.25 |
|  | Maze (Ours) | **48.43** | **39.17** | **81.88** | **53.90** |
| Qwen3-4B | Original | 49.11 | 36.50 | 89.86 | 84.46 |
|  | only last layer | 53.0 | 40.8 | 91.0 | 86.34 |
|  | only middle layer | 53.2 | 41.1 | 91.6 | 86.68 |
|  | w/o Cross-layer metrics | 53.5 | 42.4 | 92.1 | 86.46 |
|  | w/o Cross-layer fusions | 52.5 | 42.7 | 92.0 | 86.10 |
|  | random pairs | 52.3 | 40.7 | 91.5 | 85.14 |
|  | Maze (Ours) | **54.13** | **43.18** | **92.67** | **87.28** |

## B.2    VISUALIZATION

**Setup.** We project hidden state layers for a fixed query and its candidate exemplar paths in 2D using principal component analysis (PCA). Concretely, for each inspected layer $l$, we collect the average hidden state along each candidate path, fit a PCA per layer on the union, and plot the resulting 2D embeddings. Within each panel, a polyline traces the exemplar order in a candidate path; the line darkness encodes the empirical correctness of that path when used as a few-shot prompt, and the query is marked separately.

**Findings.** Across layers, we do not observe a stable, simple geometric signature that linearly separates high-accuracy from low-accuracy paths (e.g., a single cluster or monotone trajectory); rather, the arrangement and relative proximities of subpaths change with depth. This suggests that naive heuristics based on a single layer or purely local distances are insufficient and motivate the complex cross-layer metric learning used by our Maze.

**Interpretation and caveats.** We use PCA for its linearity and interpretability; other nonlinear techniques such as t-SNE and UMAP may reveal tighter local neighborhoods but can mislead about global structure; hence, we treat all plots as diagnostic rather than evidentiary and rely on held-out accuracy/BoN emulation for quantitative claims.

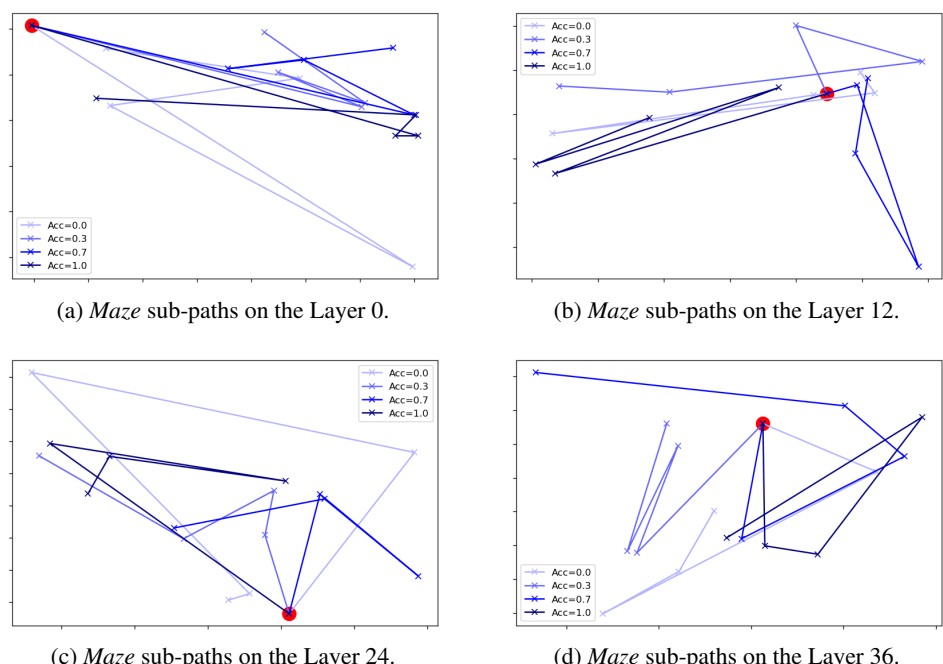

(a) *Maze* sub-paths on the Layer 0.      (b) *Maze* sub-paths on the Layer 12.

(c) *Maze* sub-paths on the Layer 24.      (d) *Maze* sub-paths on the Layer 36.

Figure 3: **Layer-wise PCA projections of exemplar sub-paths for a single MMLU-Pro query with Qwen3-4B.** Each polyline corresponds to the ordered exemplars in a candidate few-shot path. Darker lines denote paths that yield correct answers when executed as prompts. The red circle marks the test query's embedding. Together, the panels illustrate that high-accuracy paths do not admit a simple linear signature shared across layers, motivating cross-layer, order-aware metric learning.

### B.3 IMPACT OF THE NUMBER OF CONTRASTIVE PAIRS FOR TRAINING *Maze*

As expected, larger training sets generally improve performance, but gains are weakened as the number of contrastive pairs exceeds 200 thousands. In Figure 2a, when the improvement induced by *Maze* in the original LLM is greater, the effect of additional pairs becomes more volatile. For example, Qwen3-8B exhibits the largest *Maze* gains (and the strongest original performance), but shows the largest variance as the training size increases. We hypothesize that stronger base models activate richer knowledge associations under *Maze*, amplifying both benefits and sensitivity. This suggests *Maze* may not merely *organize* existing knowledge of LLM, but could also surface *new* connections; we leave a deeper investigation of this hypothesis to future work. We also observe that performance gains saturate beyond roughly 15 training epochs (see Figure 5a in Appendix B).

### B.4 EFFECT OF PATH-POOL AND MAPPING PARAMETER SCALE

In inference, we generate $M$ random candidate paths, compute their energies, and select the highest energy path as presented in Section 2.6. Calculating the energy of a single path typically takes $\mathcal{O}(10^{-3})$ seconds, negligible relative to an LLM call. Empirically, beyond a threshold like 50 as presented in Figure 2b, increasing $M$ yields diminishing returns: selecting the best of $M = 100$ paths often performs comparably with selecting from $M = 10,000$, while costing only $\sim 1\%$ as much for energy evaluation. Thus, a modest $M$ suffices in practice.

We also vary the parameter count of the metric mappings (see Figure 5b in Appendix B). A larger metric model capacity generally helps, but gains saturate beyond roughly 0.35 M, corresponding to $r = 64$ and $h = 96$. This indicates that the effectiveness of *Maze* does not depend on large mapping models; rather, it stems from the accurate modeling of LLM *inference history* as a compact and transferable structure, closer in spirit to a lightweight *world model* than a reward model. Conceptually, *Maze* offers a scaling paradigm complementary to parameter scaling.

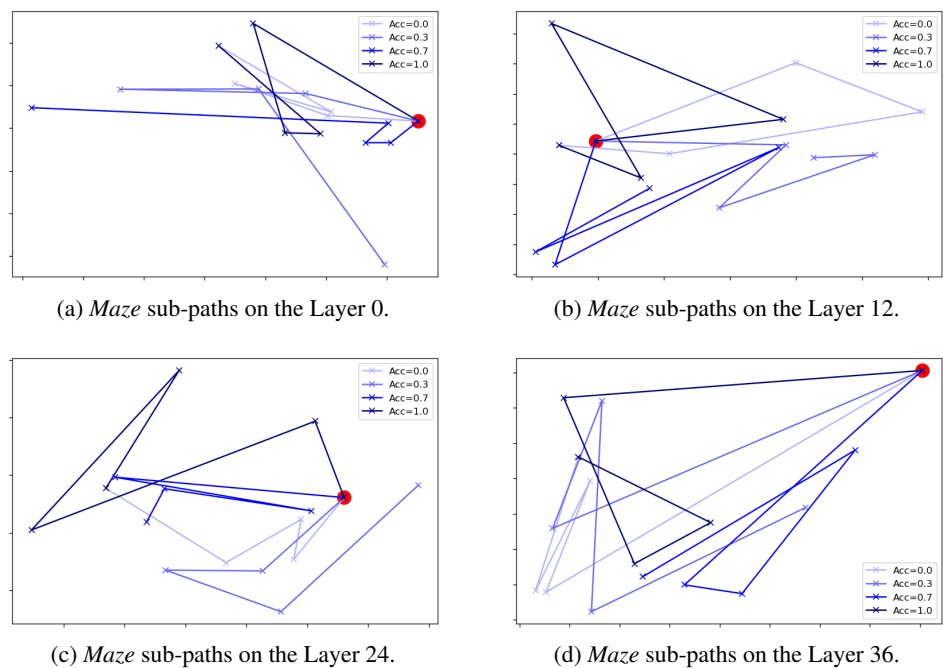

(a) *Maze* sub-paths on the Layer 0.       (b) *Maze* sub-paths on the Layer 12.

(c) *Maze* sub-paths on the Layer 24.       (d) *Maze* sub-paths on the Layer 36.

Figure 4: Same visualization protocol as Figure 4, for a GPQA test query with Qwen3-4B. We observe similar qualitative behavior: path correctness varies with fine-grained geometry and exemplar order, and salient patterns are not preserved by any single layer alone, reinforcing the need for Maze's multilayered metrics and fusion mechanism.

## B.5 GENERALITY TO UNSEEN TASKS AND DOMAINS

The main benefit of exemplars is reusable reasoning patterns rather than new knowledge. Even when the exemplar domain is different from that of the test set, as long as the exemplars contain useful intrinsic reasoning structures, like logical patterns, they can still strengthen the model in solving new problems. To validate this view, we further conduct small-scale cross-domain ablations as follows:

- Train on GSM8K, test on MATH500: +2.2 accuracy gain (vs. +2.5 under our default multi-task training setting)
- Train on MATH500, test on GSM8K: +2.6 (vs. +2.8 default)
- Train on GSM8K and MATH500, test on MMLU-pro: +4.7 (vs. +5.0 default)

Here, the reported numbers are the accuracy gains achieved by *Maze* over the original LLM; the numbers in parentheses are the gains from our default setting as reported, where *Maze* is trained on all task training sets jointly. We are initially surprised by how well the cross-domain training carries over, *i.e.*, *Maze* trained purely on GSM8K still improves performance on MATH500 to almost the same extent as a *Maze* trained directly on MATH500. This suggests that *Maze* is learning a meta-model of the LLM's own reasoning behavior, not task-specific factual content. The learned metrics appear to capture more abstract logical patterns in the model's hidden space, encouraging the LLM to focus on reasoning patterns rather than pure memorization.

## B.6 DEPLOYMENT ON BLACK-BOX LLMS

Importantly, *Maze*'s assumption of having access to hidden states is modest and common in advanced prompting methods that go beyond naive black-box use. In practice, any compatible LLM (or smaller proxy model) can serve as the encoder to provide these hidden states. In fact, we found that using a single smaller LLM as a universal encoder for all exemplars still maintains comparable performance, underscoring that *Maze* is deployable even when direct access to the weights of a large

model is limited. For instance, when we use Qwen3-4B as a shared global encoder for all the LLM (including LLaMA and Mistral) and the baselines that require encoding exemplars, the performance gain remains almost unchanged. See Table 4 for more details.

Table 4: **Accuracy gains with a swappable encoder**. We compare performance improvements (absolute accuracy gain over the raw LLM) when (i) using LLM itself to produce exemplar/query embeddings ("LLM itself") versus (ii) using a lightweight proxy encoder (Qwen3-4B). "/" indicates settings that are not applicable. Note that the proxy encoder is only applicable to Maze.

|  |  | MMLU-pro | | AIME 21-24 | |
|---|---|---|---|---|---|
| Encoder Type | | LLM itself | Qwen3-4B | LLM itself | Qwen3-4B |
| Mistral-7B | EASE | +2.5 | / | +5.9 | / |
|  | Maze (Ours) | +4.3 | +4.7 | +10.8 | +10.8 |
|  | Majority Vote | +4.7 | / | +11.4 | / |
|  | MultiMaze (Ours) | +13.2 | +12.0 | +19.6 | +19.5 |
| LLaMA3-8B | EASE | +3.7 | / | +5.2 | / |
|  | Maze (Ours) | +7.1 | +6.8 | +10.5 | +10.0 |
|  | Majority Vote | +12.3 | / | +11.5 | / |
|  | MultiMaze (Ours) | +20.9 | +20.7 | +17.3 | +17.8 |
| Qwen3-8B | EASE | +7.2 | / | +2.8 | / |
|  | Maze (Ours) | +10.3 | +10.1 | +5.7 | +5.5 |
|  | Majority Vote | +22.5 | / | +8.1 | / |
|  | MultiMaze (Ours) | +24.7 | +24.4 | +15.6 | +15.7 |
| Qwen3-14B | EASE | +4.8 | / | +1.5 | / |
|  | Maze (Ours) | +8.2 | +8.7 | +3.3 | +3.8 |
|  | Majority Vote | +12.2 | / | +3.1 | / |
|  | MultiMaze (Ours) | +17.0 | +17.3 | +9.3 | +9.3 |

This encoder also enables transfer to closed-source black-box LLMs. We cache the exemplar embeddings with a public encoder such as Qwen3-4B, externally learn *Maze*'s metric, then use that metric to select a few-shot prompt for the black-box model. Specifically, in a small pilot on GPT-5 with instant-answer and non-thinking mode using 500 random MMLU-pro test items:

- Step 1. We have already cached the exemplar embeddings from the training set with a public encoder like Qwen3-4B.
- Step 2. We sample 1k different sequences made of these training exemplars as input prompts and feed them into GPT-5 and record the performance to train *Maze*.
- Step 3. During the inference stage, we first use the encoder to encode the test query, then we use the metrics in Maze and the exemplars pool to select an optimal path that corresponds to a few-shot input prompt, and finally we use this input prompt to guide the GPT-5 to generate the answer.

As in the pipeline presented above, *Maze* has no access to the internal weights of a black-box LLM such as GPT-5. The empirical results are shown in Table 5.

Table 5: **Transfer to a closed-source LLM (GPT-5) via a proxy encoder**. *Maze* and *MultiMaze* use cached embeddings from a public encoder (Qwen3-4B) to plan few-shot prompts without accessing GPT-5 internals. Results report reasoning accuracy for single-call ($N$=1) and multiple-call settings ($N$=8), compared with raw GPT-5 and standard multi-call baselines.

|  |  | MMLU-pro | GPQA-main | AIME 21-24 |
|---|---|---|---|---|
| single-call ($N$=1) | Raw | 86.2 | 82.5 | 75.4 |
|  | Maze | 88.9 | 86.5 | 81.4 |
| Multiple-call ($N$=8) | Majority Vote | 89.4 | 86.9 | 81.2 |
|  | Oracle BoN | 91.0 | 88.2 | 81.8 |
|  | MultiMaze | 91.4 | 89.5 | 82.3 |

These empirical results clarify that the cached (via a lightweight encoder like Qwen3-4B) internal representations are used only to pre-compute the metric space with a one-time and offline cost.

### B.7 HYPERPARAMETER CHOICES (HIDDEN DIMENSION, METRIC DIMENSION, MAPPINGS)

Figure 5b illustrates how the accuracy of *Maze* on the MMLU-pro subset varies with the number of parameters, adjusted by modifying the hidden dimension or metric dimension. As stated in Appendix B.4, we observed that increasing either the metric dimension or the hidden dimension leads to a larger parameter count and improved performance. When the model size reaches approximately 0.35 million parameters, performance begins to saturate, with gains becoming more marginal. Despite that, the accuracy still continues to increase asymptotically toward the Oracle-guided Best-of-$N$ performance. Below are some example results using Qwen3-4B in MMLU-pro, where $h$ denotes the hidden dimension and $r$ the metric dimension. See Table 6 for more details.

Table 6: **Ablation on varying hidden dimension and metric dimension.**

| hidden-dim $h$ | metric-dim $r$ | #Params (M) | MMLU-pro | MATH-500 |
|---|---|---|---|---|
| 48 | 32 | 0.16 | 49.2 | 84.9 |
| 48 | 64 | 0.24 | 52.7 | 85.6 |
| 96 | 32 | 0.22 | 53.2 | 87.2 |
| 96 | 64 | 0.35 | 54.1 | 87.3 |
| 128 | 96 | 0.48 | 54.3 | 87.4 |

For practical considerations and limited computing resources, we did not scale further. However, we believe that with more data records and computational resources, *Maze* would exhibit more pronounced scaling behavior. Regarding Mass Mapping and Fusion Function, we also experimented with alternative mass mapping functions beyond the simple linear projection, as well as a deeper fusion network to replace the current shallow MLP. However, we found that these more complex variants did not yield significant improvements over our original design. For example, using an attention-based mass mapping or a 2-layer MLP as fusion mapping leads to marginal increase less than 0.5%, while increasing implementation complexity. This suggests that our chosen combination "linear projection + simple FC" strikes a good balance between simplicity and effectiveness.

### B.8 EXPERIMENTAL RIGOR: COMPUTE VS. PERFORMANCE ANALYSIS

**Compute cost scales linearly with the number of LLM calls.** *Maze* is an external planner: it selects prompts via lightweight metric evaluations on cached embeddings, so the dominant test-time cost is the number of LLM invocations. For example, if *Maze* uses 3 calls, *i.e.*, 3 candidate paths tried in parallel or sequentially, the total tokens generated are roughly 3x of a single-call's tokens. This is expected and is identical to how baselines like Best-of-$N$ or self-consistency behave when increasing $N$. The key point is that *Maze* achieves much higher accuracy in a given budget $N$, because it smartly chooses paths rather than drawing $N$ random attempts.

**No increase in output length.** *Maze* does not modify decoding or encourage longer generations. Empirically on MATH-500, the average output length is essentially unchanged (642 tokens for the raw single-call baseline vs. 647 tokens with *Maze*). In practice, the selected exemplars often streamline reasoning by improving the prompt structure rather than increasing verbosity. The model's internal generation dynamics remains unchanged, *i.e.*, *Maze* only influences which exemplars are in the prompt, not how the model autoregressively produces tokens.

In summary, *Maze* provides accuracy gains "nearly for free" in terms of per-call token cost. The LLM-call count is the major cause of inference cost, and *Maze* minimizes that count by design.

## B.9   OTHER RESULTS

**Training-set size and saturation**. Accuracy improves as the number of contrastive pairs increases and then saturates beyond $200k$ pairs; stronger base models show a larger variance with further scaling (Fig. 2a).

**Inference pool size**. Energy evaluation is matrix-vector dominated and sub-millisecond per path on our setup; accuracy plateaus after modest pool sizes (e.g., 60-80 paths) with diminishing returns toward (Fig. 2b).

**Mapping capacity**. Increasing the parameter count helps but saturates up to 0.35 millions (Appendix Fig. 5b), indicating that the benefit of Maze comes from structure, not just from large parametric capacity. A larger Maze model can lead to better performance, yet the compute-accuracy trade-off needs consideration.

**Cost–accuracy trade-off**. *MultiMaze* curves (Fig. 5c - 5d) show that 3 to 5 calls recover a substantial fraction of the performance of the majority vote of 32 LLM calls, consistent with the hypothesis that selection beats aggregation when sample errors are correlated.

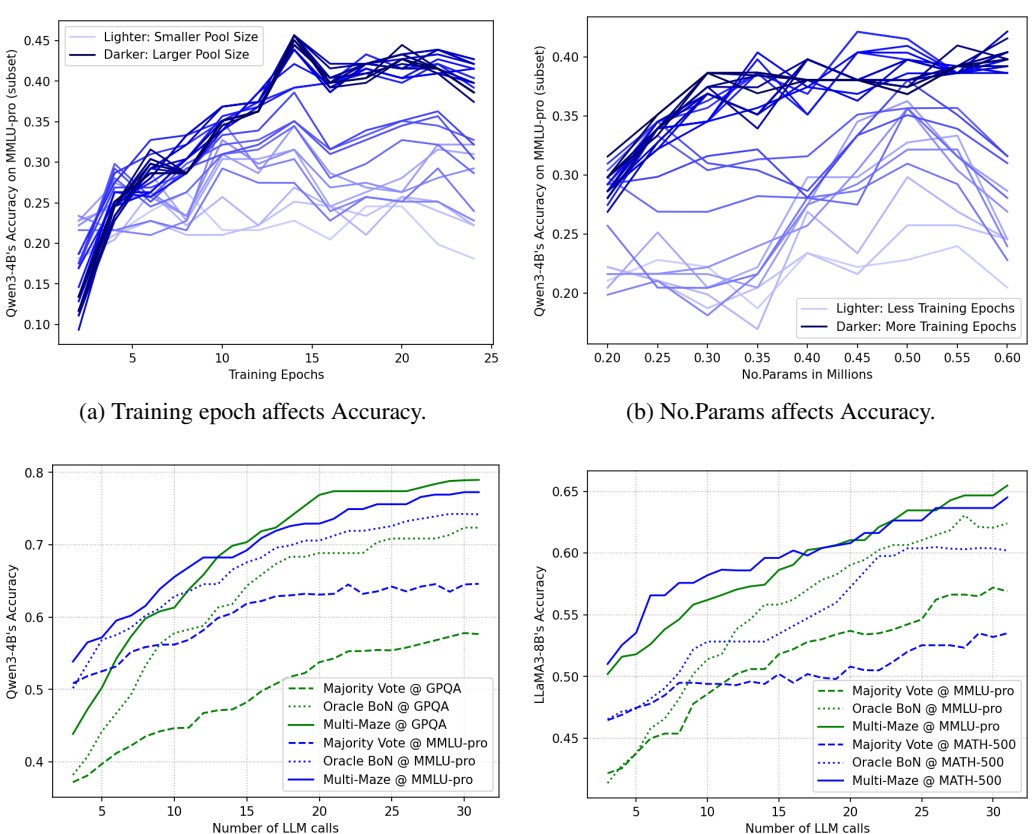

(a) Training epoch affects Accuracy.  (b) No.Params affects Accuracy.

(c) Multi-Maze vs. Majority Vote with Qwen3-4B.  (d) Multi-Maze vs. Majority Vote with LLaMA3-8B.

Figure 5: **Scaling behavior of Maze.** (**a**) Training epochs vs. accuracy on MMLU-Pro: more epochs yield steady gains that saturate after 15 epochs; darker hues indicate a larger pool size of paths (from 8 to 128). (**b**) Parameters count vs. Accuracy on MMLU-pro (Qwen3-4B): darker hues indicate a larger training epoch (from 2 to 32). (**c**) Multi-call Maze vs. Oracle-guided BoN and Majority Vote (Qwen3-4B): with the same or fewer calls, top-$k$ path selection closes most of the gap to 32-sample self-consistency. (**b**) Multi-call Maze vs. Oracle-guided BoN and Majority Vote (LLaMA3-8B): similar trend across backbones, highlighting width–depth convertibility under limited budgets.

# C  APPENDIX: MORE DISCUSSION

## C.1  WHY CAN SINGLE-CALL MAZE OUTPERFORM MAJORITY VOTE?

Consider $N$ candidate prompts with independent success probabilities for the same query. Let $p\text{vote}(N)$ denote the success probability of the majority vote under i.i.d. sampling with mean $\bar{p}$ and independence. Let $p^* = \max_i p_i$ and let Maze select the argmax with probability at least $1 - \eta$ (by Theorem 1, take $\eta$ arbitrarily small with sufficient margin).

**Proposition 1** (Selection vs. Aggregation). *If $p^* > \bar{p}$ and $\bar{p} \leq 1/2 + \gamma$ with $\gamma$ small, then for moderate $N$,*

$$\Pr[\textit{Maze correct}] \geq (1 - \eta) \cdot p^* \geq 1 - \exp(-2N\gamma^2) \geq \Pr[\textit{Majority vote correct}]. \qquad (11)$$

*In particular, when the path-quality distribution is heavy-tailed or when sample errors are positively correlated (invalidating independence), single-call selection can dominate majority aggregation.*

*Proof.* Under independence with mean $\bar{p} \leq 1/2 + \gamma$, a multiplicative Chernoff bound yields $\Pr[\text{vote correct}] \leq 1 - \exp(-2N\gamma^2)$. Maze picks a path whose success probability is within the top of the pool; by Theorem 1 and margin $\Delta$, it identifies the maximizer with probability $\geq 1 - \eta$, achieving success at least $(1 - \eta)$. If $p^*$ is significantly larger than $\bar{p}$ (e.g., due to heterogeneity of prompts), then for feasible $N$ and small $\gamma$ the stated inequality holds. With positive dependency among samples (e.g., shared error modes), the effective variance reduction of voting weakens further, tightening Maze's advantage. $\qquad\square$

**Discussion.** Majority vote improves reliability when many *independent* samples have $\bar{p} > 1/2$. In practice, samples often share failure modes (correlation), and prompt quality is highly skewed. Maze exploits this skew by *selecting* rather than *averaging*, achieving higher accuracy with lower TTC when it reliably identifies a top-quality path.

## C.2  MOTIVATION AND THEORETICAL ANALYSIS ON METHODOLOGY DESIGN

**Why metric geometry on hidden states?** Hidden-state trajectories induced by prompts and queries live in low-intrinsic-dimensional subspaces. If rewards (success probabilities) vary Lipschitz-continuously over these embeddings, then a Johnson–Lindenstrauss projection preserves relative distances with small distortion, enabling a shallow scoring network to approximate reward ordering uniformly over a large candidate pool. This justifies modeling Maze's energy with linear projections plus a small MLP, as formalized in our Theorem 1.

**asymmetric, order-aware metrics.** Empirically, in-context learning is order sensitive; different permutations can swing accuracy from near-SOTA to chance. *Maze* encodes this by using directional (non-commutative) distances and cross-layer couplings, allowing the energy to reflect both which exemplars appear and where they appear along the path. This matches prior observations of strong order effects and improves separability among high-quality prompts.

**Selection vs. aggregation.** Majority-vote and related compound systems improve reliability by averaging many samples, but recent analyses show the gains can be non-monotonic with the number of calls due to heterogeneity and correlated errors. *Maze* targets the tail of the path-quality distribution by selecting a small set of high-value paths; when independence is violated, selection can dominate aggregation at the same or lower budgets (Appendix C.1).

**Generalization of the energy.** Training on contrastive local edits provides margin-based supervision centered near actual decision boundaries. Classical generalization theory (*e.g.*, Rademacher complexity bounds for margin losses) supports that such supervision yields uniform generalization over held-out paths with sample complexity scaling $O(1/\sqrt{N})$, consistent with our empirical saturation curves.

**Complexity considerations.** With a small projection dimension and hidden width, energy evaluation over thousands of candidates is cheaper than a single LLM forward pass, enabling large path pools to be scanned under one query embedding. This aligns Maze with a planner rather than a reward model: it externalizes interaction geometry once and amortizes it across inference.

## D   RELATED WORK

**Chain-of-Thought Prompting and Few-Shot Reasoning.**  Chain-of-Thought (CoT) prompting dramatically improves LLM reasoning by including exemplar reasoning chains in the prompt, enabling models to generate intermediate steps for arithmetic, commonsense, and symbolic tasks (Wei et al., 2022). Providing just eight CoT demonstrations allows a 540B-parameter LLM to achieve state-of-the-art accuracy on the GSM8K benchmark without fine-tuning (Cobbe et al., 2021). More broadly, few-shot learning, where GPT-3 (175B) performs new tasks from a handful of text examples, highlights emergent in-context learning capabilities in large models (Brown et al., 2020). Despite these gains, CoT remains highly sensitive to the choice and order of exemplars, and single-pass decoding can still produce incorrect solutions, motivating ensemble and verification methods for greater robustness (Wang et al., 2022).

**Retrieval and Selection of In-Context Examples.**  Adaptive example selection has emerged to address CoT's sensitivity. Learning to retrieve In-Context examples trains a bi-encoder retriever via distilled LLM feedback to identify high-quality exemplars, yielding consistent improvements across 30 tasks (Wang et al., 2023). A parallel approach uses a reward model to label candidate prompts and then distills a cross-encoder retriever for efficient selection (Rubin et al., 2021). Recent surveys of retrieval-augmented in-context learning draw connections to classic information retrieval, emphasizing the importance of task-specific relevance metrics for exemplar quality (Qiao et al., 2022). However, these methods treat exemplars as unordered sets, overlooking the potential benefits of optimizing the exemplar sequence as a structured path through embedding space.

**Test-Time Scaling and Inference Strategies.**  To further enhance reliability, Test-Time Scaling (TTS) allocates extra compute at inference by sampling multiple CoT outputs and reranking them with a verifier model. Snell et al. (2024) demonstrates that, under compute-optimal TTS strategies, a 1B-parameter model can outperform a 405B-parameter model on MATH-500 and AIME24. Google's Gemini Ultra (Team et al., 2023) uses CoT@32 voting strategy to achieve state-of-the-art results on MMLU. More generally, Zhang et al. (2025) analyzes performance vs. FLOPs trade-offs across greedy search, Best-of-$N$, weighted voting, and tree-search variants, showing that smaller models with advanced inference often lie on the Pareto frontier. Reward-aware compute-optimal scaling formalizes this adaptivity, tuning hyperparameters per prompt to maximize accuracy under a fixed compute budget (Snell et al., 2024).

**Search-Based and Geometric-based Reasoning.**  Tree of Thoughts (ToT) maintains a search tree of "thought" states, coherent text segments, allowing lookahead, backtracking, and pruning to solve planning problems with substantial gains (Yao et al., 2023). Monte Carlo Tree Search (MCTS) has also been adapted for text generation by pairing policy models with learned value functions, but its rollouts incur high compute costs and diminishing returns in efficiency (Silver et al., 2016). EASE (Wu et al., 2023) frames exemplar selection as a neural-bandit-optimized ordering problem, seeking a single fixed sequence that performs well across all queries by iteratively balancing exploration and exploitation in the embedding space. Flatness-Aware Prompt Selection (Shen et al., 2023) computes a prompt flatness score by measuring the robustness of LLM loss to small perturbations in prompt tokens, thus selecting exemplars that lie in "flatter" regions of the loss landscape. P-Adapters (Newman et al., 2021) insert lightweight adapter modules into the LLM's early layers to convert natural language prompts into continuous embeddings, learning a mapping from discrete prompt text to high-quality continuous prompts.

**Mathematical Reasoning via Test-time Methods**  Beyond static prompting, recent advances exploit iterative refinement and tool integration at inference to boost mathematical accuracy. Beyond CoT (Wei et al., 2022) and its variants (Yao et al., 2023; Leang et al., 2024), self-Refine iteratively reviews and corrects model outputs without external supervision, boosting solution coherence (Madaan et al., 2023). Tool-augmented frameworks such as Toolformer (Schick et al., 2023) and MATHSENSEI (Das et al., 2024) enable LLMs to call symbolic calculators or program interpreters for exact computations (Gao et al., 2023; Chen et al., 2022). Self-Verification trains models to judge the correctness of their own reasoning steps, further reducing errors (Weng et al., 2022). Yang et al. (2024) argues that current heuristics are insufficient for robust problem solving at the theorem level, motivating algorithms that can produce single-pass verifiable proofs with minimal backtracking. Although effective, these pipelines still involve multiple reasoning passes or external calls, underscoring the need for a unified, single-pass, compute-efficient framework.

