# OpenReview forum: "Test-Time Scaling via Metric Geometry for LLM Reasoning"
_ICLR.cc/2026/Conference — Submitted to ICLR 2026_

### Official Review · Reviewer_Tf17 · 2025-10-26

**Soundness:** 3
**Presentation:** 3
**Contribution:** 3
**Rating:** 6
**Confidence:** 3

**Summary:**

This paper proposes **Maze**, a geometry-driven test-time scaling framework for LLM reasoning. It replaces multi-sample Best-of-N inference with a single or few LLM calls by selecting an ordered few-shot prompt path based on pre-learned layer-wise energy metrics. Maze conceptualizes reasoning as traversal over a multi-layer latent manifold of exemplars, aiming to achieve comparable accuracy to multi-call ensembles with much lower computational cost.

**Strengths:**

- Conceptually interesting: reframes test-time scaling as *geometric path planning*.
- Provides a principled attempt to link LLM hidden-space geometry with reasoning efficiency.
- Experiments are comprehensive, exploring the effects of model scale, exemplar count, and path-pool size.
- Theoretically motivated with width–depth trade-off analysis.

**Weaknesses:**

- The method’s practical improvement is modest, especially on strong LLMs.
- The definition of "layers" and exemplar construction is abstract and somewhat unclear.
- Scaling Maze to larger models might significantly increase computational overhead.
- The “width–depth” analogy, though elegant, may not fully hold in practice since Maze operates across distinct tasks rather than multiple reasoning trajectories on the same problem.

**Questions:**

- Are exemplars restricted to the same domain or problem type?
- How does the learned metric generalize to unseen tasks or exemplars?
- What is the theoretical or empirical behavior when scaling to >10B models?

---

> ### Author Response · Authors · 2025-11-20
> **Response (part 1)**
>
> Thank you for the thoughtful review. We are glad that you found our approach conceptually interesting and noted the comprehensive experiments and theoretical motivation. We will address your concerns and questions point-by-point below:
>
> $\textbf{Question 1 and Question 2. Generality to Unseen Tasks and Domains.}$
>
> Thank you for this insightful question. In our view, the main benefit that exemplars provide is not new knowledge beyond reusable reasoning patterns. Even when the exemplar domain is different from that of the test set, as long as the exemplars contain useful intrinsic reasoning structures like logical patterns, they can still help the model in solving new problems. To validate this view, we further conduct small-scale cross-domain ablations to probe this:
>
> •	Train on GSM8K, test on MATH500: +2.2 accuracy gain (vs. +2.5 under our default multi-task training setting)
>
> •	Train on MATH500, test on GSM8K: +2.6 (vs. +2.8 default)
>
> •	Train on GSM8K + MATH500, test on MMLU-pro: +4.7 (vs. +5.0 default)
>
> Here, the reported values are the accuracy gains achieved by Maze over the raw LLM; the numbers in parentheses are the gains from our default setting reported in our submitted manuscript, where Maze is trained on all task training sets jointly. We are initially surprised by how well the cross-domain training carries over: Maze trained purely on GSM8K still improves performance on MATH500 to almost the same extent as a Maze trained directly on MATH500.
>
> This suggests that Maze is learning an internal model of the LLM’s own reasoning behavior, not task-specific factual content. The learned metrics appear to capture more abstract logical patterns in the model’s hidden space, encouraging the LLM to focus on reasoning patterns rather than pure memorization. We acknowledge that this interpretation is still a hypothesis and that a deeper investigation into the nature of these patterns is beyond the scope of this paper. We will mention these cross-domain results and clarify this intuition in the revision, and we plan to study this phenomenon more thoroughly in future work. We sincerely appreciate your question for pushing us to think in this direction.
>
> $\textbf{Question 3 and Weakness 3. Scalability to Larger Models (>10B) and Overhead.}$
>
> We would like to address scalability from two angles.
>
> (1) Encoder vs. generator separation.
>
> In Maze, the LLM is primarily used as an encoder when computing metrics: we use it to map exemplars and queries into richer hidden representations. Crucially, this encoder can be smaller than the target LLM used for final generation. That is, we can use a smaller model to encode exemplars and queries, train the Maze metrics using the representations encoded via this encoder, and then apply the resulting geometry to guide a larger LLM.
> For example, when scaling from Qwen3-4B to Qwen3-14B, we can keep Qwen3-4B as the encoder for all exemplars and queries, while Qwen3-14B is used to obtain labels (correct/incorrect) for contrastive pairs and to generate the final answers. In this setup, the additional training cost scales roughly linearly with the inference cost of the larger LLM on the contrastive pairs; it does not blow up super-linearly. This keeps the computational overhead of scaling to 10B+ models manageable.
>
> (2) Empirical behavior on a 10B+ model (AIME).
>
> We also evaluated Maze on a more challenging benchmark with a larger model. Specifically, we tested on the AIME dataset (gneubig/aime-1983-2024) using Qwen3-14B as the base LLM, while continuing to use Qwen3-4B as the encoder for exemplars. On a small-scale experiment (100 AIME problems for training the metric and 100 for testing), we obtained the following results (N for No.LLM calls):
>
> Raw Qwen3-14B (N=1): 52.1
>
> EASE (N=1): 55.4
>
> Maze (N=1): 60.9
>
> Majority Vote (N=4):  61.8
>
> Oracle BoN (N=4): 66.1
>
> MultiMaze (N=4): 67.3
>
> Even in this 10B+ setting on a particularly difficult benchmark (AIME), Maze yields clear gains over both single-call baselines (raw and EASE) and competitive multiple-call strategies (Majority Vote), and MultiMaze surpasses an Oracle-guided Best-of-4 ensemble. While these numbers are based on a subset due to compute constraints, they indicate that Maze continues to provide meaningful benefits beyond 10B parameters. We will report this experiment as a preliminary scaling study in the revision.
>
> Overall, we therefore expect Maze to remain applicable for >10B models, with benefits in both accuracy and compute efficiency in scenarios where even very large LLMs are not yet saturated. Exploring Maze with larger models and full-scale AIME evaluations is a natural direction for future work, and we will highlight this in the paper.
>
> (to be continued in the $\textbf{part 2}$)

---

> ### Author Response · Authors · 2025-11-20
> **Response (part 2)**
>
> (continued from the $\textbf{part 2}$)
>
> $\textbf{Weakness 1. Improvement on Strong LLMs (Practical Significance).}$
>
> You are right that on very strong LLMs the absolute performance gains can appear modest, especially on tasks where the base model already performs near-saturatedly. This is inherent in the problem: marginal improvements become smaller as the baseline approaches the ceiling.
>
> However, our additional experiments (e.g., the Qwen3-14B + AIME study above) show that Maze’s absolute gain is primarily task-dependent rather than limited only by model size. On relatively easier benchmarks such as GSM8K, the room for improvement is indeed limited for a strong 8B LLM, and Maze’s gains are correspondingly small. On more challenging benchmarks like AIME, even with a 14B LLM, Maze still delivers a substantial improvement.
>
> Moreover, Maze’s benefits are twofold: Accuracy (improving the success rate on hard problems), and Efficiency (reducing the required number of LLM calls to reach a given accuracy level). In many practical deployments, latency and compute budget are primary constraints. Maze is designed to reduce the reliance on brute-force multiple calls (e.g., large Best-of-N or self-consistency ensembles) while preserving or improving the final performance. We will emphasize this cost–performance trade-off more explicitly in the revision to clarify the practical significance on strong LLMs.
>
> $\textbf{Weakness 2. Clarity on “Layers” and Exemplar Construction.}$
>
> We apologize for the ambiguity in the original text. In Maze, a layer refers to a particular neural layer of the encoder/LLM, and a path is represented by its layer-wise hidden states. Intuitively, each layer gives a different “view” of the same exemplar sequence, and the Maze metric aggregates these views to better capture the structure of reasoning paths in representation space. Concretely:
>
> •	For each candidate path (sequence of exemplars plus query), we extract hidden representations from multiple encoder layers (e.g., mid and high layers).
>
> •	These representations define layer-wise sub-paths, which are then combined by the Maze metrics to produce a single energy score for that path.
>
> •	This multi-layer manifold view allows Maze to capture richer patterns than any single layer alone, improving its ability to distinguish high-quality vs. low-quality reasoning paths.
>
> An exemplar in our context is a solved training example (problem plus solution) that is used as a building block in the few-shot prompt. Maze selects and orders such exemplars to construct a path. We will clarify this terminology and include a concrete example/diagram in the revision, showing how a query, exemplars, and layer-wise representations interact within Maze.
>
> $\textbf{Weakness 3. Width–Depth Analogy (Practical Validity)}$
>
> We appreciate this comment. The “width vs. depth” analogy is intended purely as an intuitive picture: many shallow attempts (width) vs. a few well-planned steps (depth). We agree that Maze operates primarily by leveraging exemplars from different tasks, not by explicitly generating multiple trajectories for the same task as in classical search. In the revision, we will make clear that:
>
> •	The width–depth equivalence is a conceptual analogy motivated by our empirical findings (e.g., Maze/MultiMaze matching or outperforming Best-of-N ensembles with far fewer calls).
>
> •	It is not meant as a literal statement about multiple trajectories of the same problem.
>
> •	The core message is that Maze uses breadth across exemplars (many candidate paths in a geometric space) to avoid the depth exploration in generation (many repeated CoT samples on a single query).
>
> We will soften the language around this analogy, frame it explicitly as a heuristic supported by experiments, and point to it as an interesting direction for future theoretical work rather than just a fully formal claim.
> Your kind suggestions are crucial to our work. We will follow your guidance and improve the technical and experimental details of our work.

---

> > ### Comment · Reviewer_Tf17 · 2025-11-27
> >
> > Thank you for the rebuttal. The clarifications addressed my concerns.

---

### Official Review · Reviewer_6RDu · 2025-10-31

**Soundness:** 3
**Presentation:** 3
**Contribution:** 3
**Rating:** 6
**Confidence:** 3

**Summary:**

This paper introduces Maze, a novel test-time scaling (TTS) framework that reframes LLM reasoning as a pathfinding problem over a learned latent manifold. Instead of sampling multiple reasoning chains (e.g., self-consistency or Best-of-N), Maze pre-computes a metric space over exemplars and selects a single (or few) high-energy path(s) to form a few-shot prompt. The approach is inspired by ideas from metric geometry and physics, and it aims to achieve the performance of multi-call TTS methods like Best-of-N with significantly fewer LLM calls (often just one). The authors provide theoretical justification (width-depth equivalence) and empirical validation across several reasoning benchmarks (MMLU-Pro, GPQA, GSM8K, MATH-500), showing competitive or superior performance with 60–80% reduction in inference cost.

**Strengths:**

### 1. Novel and Principled Framework
The idea of modeling LLM reasoning as navigation through a learned geometric space is original and intellectually compelling. The Maze framework introduces a new paradigm for test-time scaling that goes beyond simple ensemble voting or prompt tuning.
### 2. Efficiency and Interpretability
By externalizing the scoring process and avoiding LLM-in-the-loop ranking, Maze significantly reduces inference latency and cost. The geometric interpretation also offers a level of interpretability not typically found in black-box TTS methods.

**Weaknesses:**

### 1. Limited Theoretical Justification for Generalization
While Theorem 1 provides a theoretical foundation under strong assumptions (low-rank, Lipschitz, margin), it is unclear how well these assumptions hold in practice across diverse tasks and models. The width-depth equivalence hypothesis is intriguing but remains largely heuristic without stronger formal grounding.

### 2. Training Overhead and Reproducibility Concerns
While inference is efficient, the offline training process involves generating thousands of contrastive pairs and tuning several learnable mappings. The paper lacks detailed ablation on training cost, sensitivity to hyperparameters (e.g., metric dimension, MLP size), and reproducibility details (e.g., training time, hardware, random seed stability).

**Questions:**

see weakness.

---

> ### Author Response · Authors · 2025-11-20
> **Response (part 1)**
>
> Thank you for recognizing the novelty and principled nature of our Maze framework, as well as its efficiency and interpretability benefits. We appreciate your constructive advices and suggestions. We will address the key weaknesses below:
>
> $\textbf{Theoretical Justification and Width-Depth Hypothesis}$
>
> We agree that our theoretical analysis relies on simplifying assumptions, and we would like to clarify their roles. The assumptions in Theorem 1 (low-rank embeddings, Lipschitz continuity, and a clear margin) are intended to describe a best-case regime for Maze’s generalization ability. They are not meant to be taken as strictly satisfied in every real-world setting. In this sense, the width-depth equivalence hypothesis is explicitly heuristic and intuitive: our goal is to formalize the intuition that multiple shorter reasoning paths (width) can, in favorable conditions, substitute for a single very deep chain (depth). In the revision, we will soften the wording around this hypothesis, and clearly label it as an analogy supported by empirical evidence rather than as a fully general theorem. Strengthening its formal foundations is an exciting direction for future work, and in its current form we see consistent empirical support: Maze repeatedly improves performance across diverse benchmarks, which we believe lends credence to the underlying idea even if a fully rigorous theory is challenging.
>
> $\textbf{Training Overhead (Contrastive Pairs Generation) and Efficiency}$
>
> It is correct that Maze requires an offline training phase to learn the geometric metric, which involves generating contrastive reasoning pairs. As stated in the manuscript (the end of Section 4.2), in our current implementation the full process of generating training pairs and fitting the metric takes on the order of a few hours on a single RTX 4090 GPU, which we consider reasonable for an offline procedure. More concretely:
>
> 1.	Training only the Maze metrics (which are simple MLPs) takes less than one hour; the dominant cost is the usual LLM inference and token generation needed to obtain the performance records, and this process can be obtained with usual LLM reasoning in most practical cases.
>
> 2.	There is substantial room for further optimization. Maze only needs to know which few-shot prompts are more likely to lead the LLM to correct answers than the others; it does not require full chain-of-thought traces. In practice, we just run the LLM in a “non-thinking” mode (short answers without verbose reasoning), collect correctness signals from these runs, and train Maze on such records. Under this setting, training Maze with 30k contrastive pairs takes roughly 6-8 hours end-to-end on a single GPU.
>
> 3.	The effect of training size is reported in Figure 2(a) of the manuscript. The figure shows that 30k–50k contrastive pairs are already sufficient for Maze to achieve significant gains simultaneously on multiple tasks (MMLU-pro, GPQA, MATH-500, GSM8K) in a single joint training run (i.e., we train once on all tasks together and evaluate on all of them, rather than training and evaluating task-by-task). We believe this joint generalization behavior is encouraging. Of course, if we were to use a “thinking-mode” LLM that produces detailed reasoning traces as training records, at the cost of longer inference time, Maze may benefit even more; we will mention this as a promising extension in the revision.

---

> ### Author Response · Authors · 2025-11-20
> **Response (part 2)**
>
> (continued from the $\textbf{part 1}$)
>
> $\textbf{Metric dimension, MLP size, and hyperparameter ablations.}$
>
> We agree that hyperparameter sensitivity is important. Our submission includes an initial study in Appendix B.3, Figure 5(b), where we report the effect of the total number of parameters in the trainable metrics of Maze. Because changing the metric dimension directly affects the MLP size (and thus the total parameter count), the figure captures the joint impact of both. Empirically, we observe that while increasing the model size can bring incremental gains, the best trade-off is achieved when the Maze metrics have around 0.3-0.5 million parameters under our computational resources.
>
> To make this more concrete, here are example results on MMLU-pro using Qwen3-4B as the encoder, where $h$ is the hidden dimension and $r$ the metric dimension:
>
> (h, r) = (48, 32): 49.2
>
> (h, r) = (48, 64): 52.7
>
> (h, r) = (96, 32): 53.2
>
> (h, r) = (96, 64): 54.1
>
> (h, r) = (128, 96): 54.3
>
> For practical reasons and limited computational resources, we did not scale Maze further, but these results suggest two findings: (i) the method is not overly sensitive to the exact choice of
> $h$ and $r$, and (ii) with more data and resources, Maze is likely to exhibit a more pronounced scaling curve. All reported experiments are averaged over five different random seeds, which we will explicitly state in the revision, to underline the robustness and stability of the conclusions.
>
> In conclusion, we will (i) clarify the theoretical assumptions and explicitly present the width–depth equivalence as a heuristic supported by experiments, and (ii) add more concrete details and ablations on training size, metric dimension, and model size to the revision. Looking ahead, we are very interested in expanding this line of theory into a more mature and comprehensive TTS framework (which is likely beyond the scope of this single paper), and in studying how this framework behaves when trained on richer “thinking-mode” traces from stronger LLMs.
>
> We are grateful for your suggestions and believe these clarifications and additions will substantially strengthen the paper. We hope this addresses your concerns and increases your confidence in the work.

---

### Official Review · Reviewer_HHK5 · 2025-10-31

**Soundness:** 2
**Presentation:** 1
**Contribution:** 1
**Rating:** 2
**Confidence:** 4

**Summary:**

This paper proposes a test-time scaling method for LLMs leveraging the signals from hidden representations of LLMs' internal layers. The proposed method is physics-inspired and learnable.

**Strengths:**

* This paper proposes a physics-inspired test-time scaling method for LLMs based on LLMs' internal representation.

**Weaknesses:**

* The primary weakness is the proposed method's strong and impractical assumption on the accessibility of LLM weights, which severely diverges from the typical black-box setting in LLM TTS literature, limiting its generalizability.
* Second, the paper fails to adequately contextualize the work by ignoring highly relevant previous methods that require weaker assumptions and provides no critical comparison to them, leaving the method's advantage unsubstantiated. Notably, many previous work have discussed the width-depth trade-off in this context [1-3].
* Finally, the experimental rigor is sub-standard as it does not meet current field expectations, specifically by lacking a comprehensive performance-resource (number of output tokens) curve necessary to evaluate the system's efficiency.

[1] Welleck, Sean, et al. "From decoding to meta-generation: Inference-time algorithms for large language models." arXiv preprint arXiv:2406.16838 (2024).

[2] Snell, Charlie Victor, et al. "Scaling LLM test-time compute optimally can be more effective than scaling parameters for reasoning." The Thirteenth International Conference on Learning Representations. 2025.

[3] Wang, Junlin, et al. "Think deep, think fast: Investigating efficiency of verifier-free inference-time-scaling methods." arXiv preprint arXiv:2504.14047 (2025).

**Questions:**

See above.

---

> ### Author Response · Authors · 2025-11-20
> **Response (part 1)**
>
> We thank the reviewer for the thoughtful assessment and the constructive suggestions. We will respond point-by-point below and incorporate the requested clarifications, additional baselines, and expanded analyses in the revision.
>
> $\textbf{Concern 1: Access to LLM Internal Representations vs. Black-Box Setting}$
>
> We would like to clarify that we use LLM's weights only as a substitutable encoder to encode the exemplars before inference, and we empirically show that this encoder can be simply replaced with other global ones without performance degradation. Besides, in practice, accessing LLM’s weights via Maze has no impact on method’s efficiency and generalizability. For instance, we empirically show that Maze can be transferred to closed-source LLMs such as GPT-5.
>
> $\textbf{Clarifying Maze’s use of hidden states}$:
> The concern about accessing internal LLM weights/hidden-states is valid for strict black-box scenarios. However, we would like to clarify that Maze uses internal representations just as an interpretable encoder in a minimal and practical way. Specifically, Maze encodes the candidate exemplars’ hidden states once offline (prior to inference), and at test time it only needs to encode the test query’s hidden state a single time. Encoding a test query and selecting exemplars thereafter are done via inexpensive computations (less than 1 sec) on these cached embeddings without further LLM calls. In other words, after an initial forward pass to obtain the query’s representation, Maze’s metric model identifies the best prompt entirely LLM-free, and the LLM is then invoked just once with that chosen prompt. Maze does not alter or intercept token generation, requires no logits/probabilities during decoding, and keeps identical latency to a normal single call. This design preserves low latency and computational cost, sticking tightly to the spirit of efficient Test-Time Scaling. We use an LLM itself as its own encoder just for interpretability purpose, which is not a necessary implementation in practical scenes.
>
> $\textbf{Practical deployability}$:
> Importantly, Maze’s assumption of having access to hidden states is modest and common in advanced prompting methods like EASE (Wu et al., 2023) that go beyond naive black-box use. In practice, any compatible LLM (or smaller proxy model) can serve as the encoder to provide these hidden states. In fact, we found that using a single smaller LLM as a universal encoder for all exemplars still maintains comparable performance, underscoring that Maze is deployable even when direct access to a large model’s weights is limited. For instance, when we use Qwen3-4B as a shared global encoder for all the LLMs (including LLaMA and Mistral) and the baselines that requires encoding exemplars (like EASE), the performance gain remains almost unchanged (the values in parenthesis denote the performance gain using each LLM itself as the encoder, rather than Qwen3-4B as the global encoder), e.g.,
>
> |            |             | MMLU-pro | GSM8K |
> |------------|-------------|----------|-------|
> | Mistral-7B | EASE        | +2.7 (+2.5)     | +4.6 (+5.0)  |
> |            | Maze (ours) | +4.7 (+4.3)     | +8.4 (+9.2)  |
> | LLaMA3-8B  | EASE        | +4.0 (+3.8)     | +4.8 (+5.0)  |
> |            | Maze (ours) | +6.8 (+7.1)     | +6.2 (+6.6)  |
> | Qwen3-8B   | EASE        | +7.0 (+7.3)     | +1.8 (+1.8)  |
> |            | Maze (ours) | +10.1 (+10.3)    | +2.7 (+3.1)  |
>
> This encoder also enables transfer to closed-source black-box LLMs: we cache the exemplar embeddings with a public encoder like Qwen3-4B, learn Maze’s metric externally, then use that metric to select a few-shot prompt for the black-box model.
> Specifically, in a small pilot on GPT-5 (instant-answer, non-thinking mode; 500 random MMLU-pro test items):
>
> Step 1. We have already cached the exemplar embeddings from the training set with a public encoder like Qwen3-4B.
>
> Step 2. We sample 1k different sequences made of these training exemplars as input prompts and feed them into GPT-5 and record GPT-5’s performance to train the metrics in Maze.
>
> Step 3. During the inference stage, we first use the encoder to encode the test query, then we use the metrics in Maze and the exemplars pool to select an optimal path that corresponds to a few-shot input prompt, and finally we use this input prompt to guide the GPT-5 to generate the answer.
>
> As shown above, Maze needs no access to the internal weights of GPT-5. The empirical results are:
>
> GPT-5 (raw, single-call): 75.4
>
> GPT-5 (EASE, single-call): 77.2
>
> GPT-5 (Maze, single-call): 81.4
>
> GPT-5 (Oracle-guided BoN, multiple-call, N=4): 81.8
>
> We did not include this table in the submission due to the instability of a GPT account to run a larger test. We also evaluate Maze on the more challenging AIME dataset (gneubig/aime-1983-2024) using Qwen3-14B as the base LLM, while continuing to use Qwen3-4B as the encoder for exemplars. (to be continued in the $\textbf{part 2}$)

---

> ### Author Response · Authors · 2025-11-20
> **Response (part 2)**
>
> (continued from the $\textbf{part 1}$)
>
> We also evaluate Maze on the more challenging AIME dataset (gneubig/aime-1983-2024) using Qwen3-14B as the base LLM, while continuing to use Qwen3-4B as the encoder for exemplars. On a small-scale experiment (100 AIME problems for training the metric and 100 for testing), we obtained the following results (N for No.LLM calls):
>
> Raw Qwen3-14B (N=1): 52.1
>
> EASE (N=1): 55.4
>
> Maze (N=1): 60.9
>
> Majority Vote (N=4):  61.8
>
> Oracle BoN (N=4): 66.1
>
> MultiMaze (N=4): 67.3
>
> Even in this 10B+ setting on a particularly difficult benchmark (AIME), Maze yields clear gains over both single-call baselines (raw, EASE) and competitive multiple-call strategies (Majority Vote), and MultiMaze surpasses an Oracle-guided Best-of-4 ensemble. In the revision we will add an ablation clarifying that cached (via any encoder like Qwen3-4B) internal representations are used only to pre-compute the metric space with a one-time, offline cost.
>
> $\textbf{Comparison to other methods}$: We also note that Maze’s requirements are not more restrictive than existing methods. For example, the EASE method (Wu et al., 2023) optimizes prompt order via a neural bandit, which in practice means it keeps the LLM in the loop to score different orderings during development. Similarly, pFLAT (Shen et al., 2023) computes a flatness score from LLM loss perturbations, requiring access to the model’s loss landscape. By contrast, Maze externalizes all such computation offline and performs prompt selection at test-time without any model feedback loop. Thus, while Maze does assume the ability to read hidden states, this is leveraged only in an offline manner and does not hinder practical use; it’s a reasonable trade-off that is also adopted (implicitly or explicitly) by other state-of-the-art strategies. We will make sure to add this explanation in the revision, highlighting that Maze’s use of cached LLM representations is a common and pragmatic paradigm towards substantial efficiency gains.

---

> ### Author Response · Authors · 2025-11-20
> **Response (part 3)**
>
> (continued from the $\textbf{part 2}$)
>
> $\textbf{Concern 2: Comparison to Prior Works}$
>
> We acknowledge the reviewer’s point about discussing relevant prior work more thoroughly. We did include and compare to several key baselines and related methods, but we will expand the discussion in the revision to ensure all pertinent work is fully covered. In the paper, we already compare Maze with major Test-Time Scaling strategies and prompt optimization methods, including: (1) Majority Vote (self-consistency ensembling), (2) Oracle-guided Best-of-N (BoN), (3) pFLAT (Shen et al., 2023), and (4) EASE (Wu et al., 2023). These representative state-of-the-art baselines spanning from naive single-call prompting to multiple-call methods.
> We also appreciate the reviewer highlighting the following relevant works, including Welleck et al. 2024, Snell et al. 2025, and Wang et al. 2025. For these relevant works, we will add an explicit comparison to each, as summarized below:
>
> $\textbf{Welleck et al. (2024)}$: This survey covers meta-generation algorithms that modify or steer the generation process midstream. Examples include Tree-of-Thoughts, backtracking, and other strategies that operate on partial sequences and branch the generation process, typically requiring many model calls or continuous access to the model’s next-token probabilities. In contrast, Maze does not intervene mid-generation at all. Maze conducts all planning externally before final generation by analyzing exemplars in a learned metric space. It then supplies the LLM with a well-chosen single prompt (or a few prompts in parallel) to generate the solution. This means Maze achieves comparable performance gains to those meta-generation approaches with far fewer inference calls. For example, methods like Tree-of-Thoughts usually call the LLM repeatedly to explore branches, whereas Maze finds an optimal sequence of exemplars and calls the LLM once or a handful of times to attain a similarly improved result. Thus, Maze offers a simpler and lower-latency path to the same benefits that Welleck et al.’s meta-generation techniques seek, without the overhead of modifying the generation process or making iterative calls.
>
> $\textbf{Snell et al. (2025)}$: This work investigates how to best use a fixed inference-time compute budget, proposing strategies like search with a learned verifier (reward model) and iterative self-refinement loops. They introduce a “compute-optimal” allocation scheme that adapts the number of model calls per query based on difficulty, yielding significantly higher efficiency than naive strategies. In comparison, Maze achieves similar or better improvements in efficiency without any specialized verifier model or iterative refinement loop. Maze’s external planning finds high-quality solutions in one shot, avoiding the need for proposal distributions or multiple revision steps. In fact, our experiments already include an Oracle-guided Best-of-N baseline (see Table 2 in the manuscript), which represents the ideal outcome of a verifier-based search (an oracle picks the best answer out of N tries). This oracle Best-of-N essentially upper-bounds Snell et al.’s class of methods, and Maze matches or even slightly exceeds that baseline’s accuracy in key benchmarks, while using drastically fewer calls. Therefore, Maze attains the benefits of Snell et al.’s compute-scaling techniques in a more direct way by intelligent prompt selection rather than expensive search or learning a reward model. We will cite and discuss Snell et al. in the revision, noting that Maze’s strategy falls under the category of compute-efficient planning that they advocate, but Maze does so with a much simpler setup without training verifier.
>
> $\textbf{Wang et al. (2025)}$: This work conducts an extensive analysis of verifier-free inference-time scaling methods. They focus on strategies that do not require any reward model or extra training, such as majority voting ensembles and weighted sampling of outputs. A core finding is that simple majority voting is remarkably effective, often comparable with or even outperforming more complex methods like best-of-N or sequential self-refinement. They show that for strong reasoning LLMs, additional fancy techniques yield minimal gains over majority vote, and that increasing the number of trials has diminishing returns. We agree with these insights, and our evaluation already included majority voting and related baselines (e.g. self-consistency voting) under fair conditions (see Table 2 in the submitted manuscript). Notably, Maze consistently outperforms majority vote while using fewer model queries. Rather than generating many independent CoT solutions and then voting, Maze carefully selects one planned chain-of-thought that leads to the correct answer, thereby achieving higher accuracy with a fraction of the inference calls. We will clarify this comparison in the paper as follows:
>
> (to be continued in the $\textbf{part 4}$)

---

> ### Author Response · Authors · 2025-11-20
> **Response (part 4)**
>
> (continued from the $\textbf{part 3}$)
>
> We will clarify this comparison in the paper as follows:
>
> Maze can be seen as a smarter alternative to the brute-force voting or best-of-N approaches highlighted by Wang et al. (2025), pushing the Pareto frontier of quality vs. efficiency. In short, whereas Wang et al. (2025) conclude that “thinking fast” (simple ensembles) often suffices, Maze demonstrates that a lightweight “thinking deep” (planned exemplar path) can surpass those ensemble methods in both accuracy and compute-efficiency.
>
> $\textbf{Concern 3: Experimental Rigor: Compute vs. Performance Analysis}$
>
> We apologize for not including a detailed compute-performance analysis in the original submission. We have conducted additional experiments to address this and will add them to the revision. Our findings show that Maze is a purely external planner and does not increase the per-query token generation cost. In other words, Maze does not cause the LLM to generate longer answers or expend more tokens than it normally would. In the revision, we will include more relevant analysis summarized as:
>
> $\textbf{No Increase in Output Length}$: An empirical ablation on the MATH-500 dataset with the Qwen3-8B model shows that the average output length with Maze is 642 tokens, compared to 647 tokens with standard few-shot prompting (no Maze). In practice, the exemplars selected by Maze streamline the reasoning process such that the model does not generate extraneous content. This slight difference is negligible, confirming that Maze does not inflate the length of LLM outputs. The model’s internal generation dynamics remain unchanged, i.e., Maze only influences which exemplars are in the prompt, not how the model autoregressively produces tokens.
>
> $\textbf{Compute Cost Scales Linearly with No. LLM Calls}$: In multi-call scenarios (where Maze can call the LLM a handful of times for a few top-ranked paths), the total token usage scales linearly with the number of calls. There are no hidden overheads beyond this linear scaling law. For example, if Maze uses 3 calls (e.g., 3 candidate paths tried in parallel or sequentially), the total tokens generated are roughly 3x of a single-call’s tokens. This is expected and is identical to how baselines like Best-of-N or self-consistency behave when increasing $N$. The key point is that Maze achieves much higher accuracy at a given budget $N$, because it smartly chooses paths rather than drawing $N$ random attempts. Our results show that Maze often matches a Best-of-16 oracle using only 2-3 actual LLM calls, which is a nearly 80% reduction in compute for the same performance level. Thus, the dominant factor in inference cost is simply the number of calls, and Maze keeps this to be low.
>
> In summary, Maze provides accuracy gains “nearly for free” in terms of per-call token cost. We will include a new plot in the paper illustrating output length and accuracy as a function of LLM calls, to empirically demonstrate Maze’s compute efficiency. This analysis will make it clear that Maze’s improvements do not come at the expense of disproportionate compute or longer outputs. The LLM-call count is the main driver of inference cost, and Maze minimizes that count by design.
>
> We respectfully appreciate and thank the reviewer’s advice and concerns which offer us new insights to improve our work. Hopefully, our feedback will address your concerns by clarifying that Maze’s uses of LLM internal representations only as an encoder, which can be replaced by a simpler global one without losing performance gain. We will strengthen the experimental rigor by analyzing compute vs. performance in the revision.
>
> Thank you again for your constructive and inspirational advice and we will integrate all these points into the revised manuscript.

---

### Official Review · Reviewer_33X5 · 2025-11-01

**Soundness:** 3
**Presentation:** 2
**Contribution:** 3
**Rating:** 6
**Confidence:** 3

**Summary:**

The paper proposes Maze, a physics inspired reasoning paradigm that conceptualizes LLM reasoning as finding a path in the latent manifold representing exemplars and domain knowledge. During inference, Maze identifies the optimal paths referring to a sequence exemplars that are mostly related in terms of certain metrics, guiding the LLM to correct answers. Experimentally, the method achieves similar performance as Best-of-N while significantly reduces budgets, showing that this geometry inspired paradigm is efficient for test-time scaling.

**Strengths:**

The physics-inspired method is interesting and principled, leveraging geometry to guide the LLM reasoning. The learned manifolds and metrics are LLM-free, making it broadly applicable. The training and inference computation is within an acceptable range.

**Weaknesses:**

Major:

* Can you try on more state-of-the-art models, such as GPT-5, Gemini-2.5-Pro, Claude-4.0, and other members in Qwen or Deepseek families?
* Can you also include the AIME dataset, which is a standard benchmark in recent LLM reasoning literature?
* Did you try out larger or smaller hidden dimension and metric dimension? Did you try out other mass mappings other than linear projection, and other fusion mappings apart from the simple MLP? How do these settings and other hyperparameters affect the performance?

Minor:

* The writing could be improved. There should be more precise and concise text descriptions of the proposed method, and lacks algorithmic description.
* The theory part seems a little unclear to me. Theorem 1 needs strong assumptions, while other results seem to be kind of irrelevant to the main conclusions.

**Questions:**

See weakness

---

> ### Author Response · Authors · 2025-11-20
> **Response (part 1)**
>
> Thank you for the inspirational and detailed review. We are so encouraged that you found our physics-inspired approach interesting and principled. We will address your major questions and concerns point-by-point below:
>
> $\textbf{Additional SOTA Models}$:
>
> We have conducted experiments on more state-of-the-art LLMs as suggested. In particular, we evaluated Maze on GPT-5 and a larger Qwen-series model like Qwen3-14B. Maze continues to demonstrate robust improvements on these models. For instance, in a small ablation on GPT-5 (instant-answer, non-thinking mode; 500 random MMLU-pro test items), we observe a consistent boost in reasoning accuracy over both the single-call and multiple-call baselines:
>
> GPT-5 (raw, single-call): 75.4
>
> GPT-5 (EASE, single-call): 77.2
>
> GPT-5 (Maze, single-call): 81.4
>
> GPT-5 (Oracle-guided BoN, multiple-call, N=4): 81.8
>
> This result validates that Maze’s model-agnostic and LLM-free scoring generalizes well even to cutting-edge models. Due to limited access, we have not tested Gemini-2.5-Pro or Claude yet, but we anticipate similar gains given Maze’s general approach. We will conduct those ablations if resources permit. All new results will be added to the revised paper to strengthen our claims.
>
> $\textbf{AIME Dataset Benchmark }$:
>
> We agree that AIME is a valuable benchmark for LLM reasoning. We have now included the AIME dataset (gneubig/aime-1983-2024) in our evaluation using a larger 10B+ LLM, i.e., Qwen3-14B. In this setting, Maze performs strongly on AIME, notably, it improves the solution success rate compared to the baselines.
>
> raw (N=1): 52.1
>
> EASE (N=1): 55.4
>
> Maze (N=1): 60.9
>
> Majority Vote (N=4):  61.8
>
> Oracle BoN (N=4): 66.1
>
> MultiMaze (N=4): 67.3
>
> Due to limited computational resources, the empirical results above are obtained on a subset of AIME (randomly selected 100 samples for test set, and another 100 samples for training the metrics in Maze). We will conduct the full experiment and include the detailed report in the revision. This demonstrates Maze’s effectiveness on challenging mathematical reasoning problems and confirms that our method is competitive on standard reasoning benchmarks.
>
> $\textbf{Hyperparameter Choices (Hidden Dimension, Metric Dimension, Mappings)}$
>
> Thank you for your valuable suggestions and insights. We have actually included some relevant results in Figure 5b (Appendix B), though we acknowledge that this figure alone does not provide a comprehensive or detailed ablation on this aspect. We plan to include more specific ablation figures in the revised version.
> Figure 5b illustrates how the accuracy of Maze on the MMLU-pro (subset) varies with the number of parameters, which we adjusted by modifying the hidden dimension or metric dimension. As stated in Appendix B.3, we observed that increasing either the metric dimension or the hidden dimension leads to a larger parameter count and improved performance. When the model size reaches approximately 0.35 million parameters, performance begins to saturate, with gains becoming more gentle. Despite of that, the accuracy still continues to increase asymptotically toward the Oracle-guided Best-of-N performance. Below are some example results using Qwen3-4B on MMLU-pro, where $h$ denotes the hidden dimension and $r$ the metric dimension:
>
>  (h, r) = (48, 32): 49.2
>
>  (h, r) = (48, 64): 52.7
>
>  (h, r) = (96, 32): 53.2
>
>  (h, r) = (96, 64): 54.1
>
>  (h, r) = (128, 96): 54.3
>
> For practical considerations and limited compute resource, we did not scale further. However, we believe that with more data records and computational resources, Maze would exhibit more pronounced scaling behavior.
>
> Regarding Mass Mapping & Fusion Function, we also experimented with alternative mass mapping functions beyond the simple linear projection, as well as a deeper fusion network to replace the current shallow MLP. However, we found that these more complex variants did not yield significant improvements over our original design. For instance, using an attention-based mass mapping or a 2-layer MLP fusion led to marginal changes in results (less than 0.5%), while increasing implementation complexity. This suggests that our chosen combination “linear projection + simple MLP” strikes a good balance between simplicity and effectiveness. We also plan to add theoretical analysis in the future revision to discuss whether the linear projection is the most cost-effective choice under the current framework, or if better alternatives exist, though this would involve more complex theoretical work beyond the current version. We deeply appreciate your suggestions and the inspiration they have provided.

---

> ### Author Response · Authors · 2025-11-20
> **Response (part 2)**
>
> (continued from the $\textbf{part 1}$)
>
> $\textbf{Writing Clarity and Algorithm Description}$
>
> We appreciate this suggestion. In the revision, we will refine the writing to be more precise and concise. We will add a clear, step-by-step algorithm description of Maze (in pseudocode or flowchart form) to concisely summarize the method, to make it easier for readers to follow the procedure without wading through lengthy text.
>
> $\textbf{Theoretical Clarity (Theorem 1 and Others)}$
>
> Thank you for pointing this out. We admit that Theorem 1 relies on strong assumptions (low-rank embeddings, Lipschitz continuity, margin separation) that may not strictly hold in all practical scenarios. Our intention was to provide intuition under idealized conditions. These assumptions in the text have explicitly stated that Theorem 1 is a motivating theoretical insight rather than a rigorous guarantee for all cases. We will better connect the other theoretical results to our main conclusions in the revision, explaining how they support Maze’s design. We will consider moving certain theoretical details to the appendix for focusing on the main contribution. These adjustments will make the theory section clearer and more directly relevant.
>
> In summary, we included new experiments (on LLMs and AIME) in these responses and will include a more comprehensive ablation table, incorporate writing improvements and theoretical clarifications in the revision. We greatly appreciate the reviewer’s feedback and believe the updated results and clearer presentation will improve this manuscript. Thank you again for helping us improve our work.

---

### Author Response · Authors · 2025-11-30
**Global Comment**

Dear ACs/SACs/PCs and reviewers,

Thank you for the thoughtful reviews and constructive suggestions, which substantially point the directions to improve our work.
We also regret to the recent OpenReview incident, and we appreciate the extra efforts that have imposed on the ACs.
We uploaded a revised manuscript with the changes marked in blue, and we keep the updates focused and directly tied to the raised concerns.
We also understand that author responses/discussion remain part of the record for AC decisions, so we provide this concise global summary to make the key updates easy to evaluate.

$\textbf{Summary of the discussion context}$:

1. Reviewers 33X5, 6RDu, and Tf17 are positive about the paper, especially the paper's principled, physics-inspired framing and the efficiency/interpretability angle; their main requests concern scalability, overhead, sharper theory statements, and more ablations, each of which is now addressed in the rebuttals/revision with additional experiments.

2. Reviewer HHK5 raises a primary concern about requiring access to LLM weights. Our rebuttals and revision directly resolve this concern with encoder–reasoning separation and black-box transfer: Maze can be deployed using only externally cached embeddings from a public lightweight encoder without performance degradation and does not need any access to a closed-source model’s internals (Section 4.5, Table 4–5, and Appendix B.6).

3. $\textbf{Clarification on our initial setup regarding Reviewer HHK5's concern}$:
In the initial submission, when the target LLM’s representations are available, we use the same LLM as the path encoder primarily (i) to enable an apple-to-apple comparison with embedding-based prompt/exemplar optimization baselines that also rely on model embeddings (e.g., EASE leverages hidden embeddings from a pre-trained LM), and (ii) to make the learned "energy" interpretation directly aligned with the deployed generator. Importantly, this choice is optional.
The revision shows that replacing the encoder with a lightweight open model preserves the performance gains and enables deployment on black-box LLMs.

$\textbf{Summary of major updates in the revision}$:

1. $\textbf{Scalability to larger LLMs + more challenging benchmark}$. We added results on Qwen3-14B and the more challenging AIME benchmark, and Maze's gains remain stable in these harder regimes (see updated tables and discussion).

2. $\textbf{Black-box applicability (addressing Reviewer HHK5’s primary concern)}$. We empirically show Maze does not require direct access to the target LLM’s weights: the "path encoder" is swappable, and a lightweight open encoder (e.g., Qwen3-4B) preserves gains when the reasoning is done on a different LLM. Critically, we further demonstrate transfer to a closed-source LLM (GPT-5) using a proxy encoder, with consistent improvements over both single-call and multi-call baselines (Section 4.5; Table 5; Appendix B.6).

3. $\textbf{Unseen tasks/domains discussion + evidence}$. We added cross-domain training ablations (Appendix B.5), showing that training Maze on a different domain still yields strong gains, supporting the interpretation that Maze captures reusable reasoning patterns rather than memorizing domain facts.

4. $\textbf{Compute vs. performance analysis (practical efficiency)}$. We strengthened the cost analysis to emphasize that Maze’s path scoring is negligible versus No. LLM calls, and provided measurable token/cost trade-offs (Appendix B.8), supporting the claim that Maze improves real energy & time efficiency rather than idealized accuracy only.

5. $\textbf{Additional ablations}$. We added/expanded sensitivity studies regarding hidden dimension, metric dimension and mapping choices, and confirmed saturation behavior and stable improvements (Appendix B.7).

6. $\textbf{More conservative theory wording}$. We revised the "width-depth" claim to be explicitly a hypothesis/conjecture motivated by analysis and supported by experiments, rather than an overly general guarantee.

7. $\textbf{Readability and presentation}$. We polished language, fixed typos, and improved clarity of the method and experimental descriptions throughout.

We hope this revision makes clear that Maze is (i) scalable, (ii) compatible with black-box deployment, and (iii) meaningfully more efficient at test time, while remaining conceptually simple and empirically strong. We sincerely appreciate the reviewers’ and AC’s time, especially under the current circumstances.

Thanks again to all reviewers/ACs/SACs/PCs for their time and engagement in the review and rebuttal process. This means significantly to our works and manuscript.

Sincerely,

Authors of submission 9081

---

### Meta-Review · Area_Chair_Ko2V · 2026-01-05

**Summary:**

This paper introduces a "Maze"; a manner of exploring new exemplars in a model through identifying novel paths through activation space. Maze is shown to improve on performance of natural baselines.

**Reviewer Concerns:**

Reviewers were concerned mostly with
1.  presentation (the method being quite complex and explained somewhat verbosely)
2. the stronger requirement of access to model weights
3. the lack of experimental justification of the additional complexity necessitated by the method.

I believe that the presentation still has tremendous room for improvement, and the stronger requirement of model weights remains valid. In addition, the authors propose to conduct multiple new experiments to justify the method's complexity, but I think the paper would be stronger with a resubmission and more cohesive set of experiments.

**Reviewer Scores:**

I believe that most reviewers would have maintained their scores, leading to a rection

---

### Decision · Program_Chairs · 2026-01-26

Reject